# LWD–TCP complex activates the morning gene CCA1 in *Arabidopsis*

Jing-Fen Wu[1,*], Huang-Lung Tsai[1,*], Ignasius Joanito[2,3,4,5], Yi-Chen Wu[1], Chin-Wen Chang[1], Yi-Hang Li[1], Ying Wang[1], Jong Chan Hong[6,7], Jhih-Wei Chu[3,4,5], Chao-Ping Hsu[2,8] & Shu-Hsing Wu[1,8]

A double-negative feedback loop formed by the morning genes *CIRCADIAN CLOCK ASSOCIATED1* (*CCA1*)/*LATE ELONGATED HYPOCOTYL* (*LHY*) and the evening gene *TIMING OF CAB EXPRESSION1* (*TOC1*) contributes to regulation of the circadian clock in *Arabidopsis*. A 24-h circadian cycle starts with the peak expression of *CCA1* at dawn. Although *CCA1* is targeted by multiple transcriptional repressors, including PSEUDO-RESPONSE REGULATOR9 (PRR9), PRR7, PRR5 and CCA1 HIKING EXPEDITION (CHE), activators of *CCA1* remain elusive. Here we use mathematical modelling to infer a co-activator role for LIGHT-REGULATED WD1 (LWD1) in *CCA1* expression. We show that the TEOSINTE BRANCHED 1-CYCLOIDEA-PCF20 (TCP20) and TCP22 proteins act as LWD-interacting transcriptional activators. The concomitant binding of LWD1 and TCP20/TCP22 to the TCP-binding site in the *CCA1* promoter activates *CCA1*. Our study reveals activators of the morning gene *CCA1* and provides an action mechanism that ensures elevated expression of *CCA1* at dawn to sustain a robust clock.

[1] Institute of Plant and Microbial Biology, Academia Sinica, Taipei 11529, Taiwan. [2] Institute of Chemistry, Academia Sinica, Taipei 11529, Taiwan. [3] Department of Biological Science and Technology, National Chiao Tung University, Hsinchu 300, Taiwan. [4] Bioinformatics Program, Taiwan International Graduate Program, Academia Sinica, Taipei 115, Taiwan. [5] Institute of Bioinformatics and System Biology, National Chiao Tung University, Hsinchu 300, Taiwan. [6] Division of Life Science, Applied Life Science (BK21 Plus Program), Plant Molecular Biology and Biotechnology Research Center, Gyeongsang National University, Jinju 660-701, Korea. [7] Division of Plant Sciences, University of Missouri, Columbia, South Carolina MO 65211-7310, USA. [8] Genome and Systems Biology Degree Program, National Taiwan University, Taipei 106, Taiwan. * These authors contributed equally to this work. Correspondence and requests for materials should be addressed to S.-H.W. (email: shuwu@gate.sinica.edu.tw).

The operation of the circadian clock allows organisms to synchronize the internal biological activities with the external 24-h rhythmic light/dark or temperature cues. Plants with a defective circadian clock have reduced fitness[1]. The circadian clock in *Arabidopsis* is mainly driven by a double-negative feedback loop formed by the morning-phased *CIRCADIAN CLOCK ASSOCIATED1* (*CCA1*)/*LATE ELONGATED HYPOCOTYL* (*LHY*) and the evening-phased *TIMING OF CAB EXPRESSION1* (*TOC1*) (refs 2–5). In the current model of the *Arabidopsis* circadian clock, the key driving force for the daily oscillation is based on the repressilatory machinery that uses a repressor of a repressor as an activator to increase the expression of clock genes[6,7]. In this model, CCA1 is a transcriptional repressor functionally overlapping with LHY for suppressing *TOC1* in the morning[2]. Reciprocally, TOC1 binds to the *CCA1* promoter to suppress *CCA1* expression in the evening[8,9]. In addition to TOC1, multiple repressors including PSEUDO-RESPONSE REGULATOR9 (PRR9), PRR7, PRR5 and CCA1 HIKING EXPEDITION (CHE) suppress *CCA1* at different times of the day[10–12].

Although a repressilator-based system can oscillate, the robustness could be enhanced by the integration of a positive feedback loop[13]. One such positive-feedback loop in the *Arabidopsis* circadian clock is formed by LIGHT-REGULATED WD1 (LWD1) and PRR9 (ref. 14). More recently, an activator and co-activator system consisting of REVEILLE8 (RVE8) and NIGHT LIGHT-INDUCIBLE AND CLOCK-REGULATEDs (LNKs) could directly target the promoters of evening genes, *PRR5* and *TOC1*, to activate their expression[15]. However, although the light-activated expression of *CCA1* prompts the synchronization of an oscillating clock, activators for *CCA1* expression at dawn are still unknown.

Here, our mathematical models show that the simultaneous activation of *PRR9* and *CCA1* by LWD1 in the clock system allows for more precise predictions of clock behaviours in both wild-type plants and clock mutants. We further demonstrated that by direct interaction with two transcription factors, TEOSINTE BRANCHED 1-CYCLOIDEA-PCF20 (TCP20) and TCP22, LWD1 could associate with the *CCA1* promoter *in vivo*. TCP20 and TCP22 are new clock proteins capable of binding to the TCP-binding site (TBS) in the *CCA1* promoter. Without LWDs, the binding of TCP20 and TCP22 to *CCA1* promoter fails to activate *CCA1*. This study first discloses an activator–coactivator complex triggering *CCA1* expression in the morning.

## Results

**The dual activation of *PRR9* and *CCA1* by LWD1.** LWD1 can activate *PRR9* (ref. 14), one of the transcriptional repressors of *CCA1* (ref. 10). However, in contrast to the increased *CCA1* level one would expect in the *lwd1 lwd2* double mutant with reduced *PRR9* expression, *CCA1* expression is also much reduced in *lwd1 lwd2* (ref. 14). We previously showed that LWD1 can directly target promoters of multiple clock genes, including a weak association with *CCA1* at ZT0 (ref. 14). This finding led us to hypothesize that, in addition to activating *PRR9*, LWD1 can also activate *CCA1*. To determine the plausibility of such interaction, we used mathematical models to examine clock performance/robustness when LWD1 activates only *PRR9* (Model I) or both *PRR9* and *CCA1* (Model II) (Fig. 1a). Among $2.4 \times 10^8$ random parameter sets examined, 1,004 met our criteria of producing oscillation in both the wild type and *lwd1 lwd2* mutant (Supplementary Note 1) for Model II as compared with only 27 of $3 \times 10^8$ random parameter sets for Model I. Thus, despite having two more parameters to search for in Model II (Supplementary Note 1), the chance of obtaining parameter sets that met our

criteria was significantly greater when LWD1 activates both *PRR9* and *CCA1*. Also, only in the circadian system with LWD1 activating both *PRR9* and *CCA1*, but not just *PRR9*, could the simulation reveal short period lengths of *cca1* and *toc1* mutants and the long period length of *prr9* mutant (Fig. 1a). Including the activation of *CCA1* by LWD1 increases the plausibility and robustness of the clock oscillation.

To account for the possible limitations inherent to these simplified models, we also assessed the impact of LWD1 on *PRR9* alone or both *PRR9* and *CCA1* by integrating the regulation into a more comprehensive and established model. We chose the model by Pokhilko *et al.*[16], involving many known clock genes, which can replicate the multiple genetic perturbations observed in the

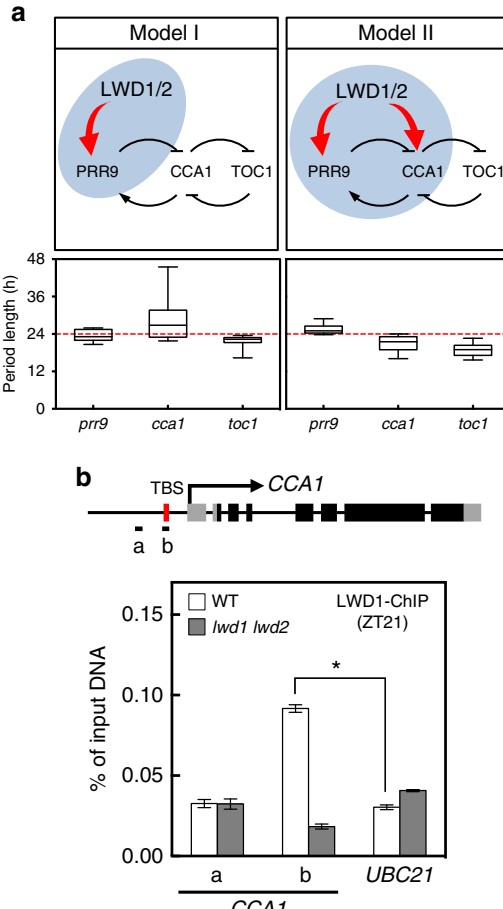

**Figure 1 | LWD1 functions as an activator in the circadian clock.**
(**a**) Proposed models for the role of LWD1 in activating the *Arabidopsis* clock system via *PRR9* (Model I) or both *PRR9* and *CCA1* (Model II). The box plots show period lengths simulated under the representative genetic perturbations (*prr9*, *cca1* or *toc1*) in these two models, with the wild-type model period being 24 h. The period lengths were calculated from a good majority of the parameter sets obtained (See Supplementary Note 1 for details). Horizontal lines are medians, box edges are interquartile range and whiskers are minimum and maximum. (**b**) LWD1 associates with the *CCA1* promoter *in vivo*. Diagram shows the gene structure of *CCA1*. Transcriptional start and exons are marked with arrow and boxes. Black boxes illustrate the coding region. Amplicons 'a' and 'b' for ChIP-qPCR assays are marked by horizontal black bars. ChIP assays involved use of the anti-LWD1 antibody at ZT21. *UBC21* was a negative control. Data are mean ± s.d. (*n* = 3 technical replicates). Asterisk indicates that LWD1 preferentially binds with the amplicon 'b' in the wild type (Student's *t* test; \**P* < 0.001). Similar results were observed in three independent experiments.

experimental results. This model has also been extended to simulate and interpret the input[17] and output pathways[18]. The sole activation of *PRR9* by LWD1 in the implemented model failed to simulate the clock defects observed in the *lwd1 lwd2* mutant (Supplementary Fig. 1a). Only the model with LWD1 activating both *PRR9* and *CCA1* gave clock features resembling the short period and low amplitude of multiple clock genes in *lwd1 lwd2* (ref. 14; Supplementary Fig. 1b). We also tested whether our modifications changed the general behaviours of the clock by adding *cca1 lhy*, *toc1* or *TOC1ox* perturbations according to the settings in the original paper[16]. The addition of LWD1 activating both *PRR9* and *CCA1* in this model also maintained the perturbed clock behaviours similar to those reported previously[16]. These behaviours included the advanced phases of *TOC1/EVENING COMPLEX* and elevated levels of *TOC1* mRNA in the *cca1 lhy* mutant (left panel in Supplementary Fig. 1c), and increased and decreased *CCA1/LHY* levels in *toc1* and in *TOC1ox*, respectively (right panel in Supplementary Fig. 1c). Therefore, the integrity of the Pokhilko model could be maintained after incorporating LWD1 as an activator of both *PRR9* and *CCA1*.

Our mathematical simulation provided a new hypothesis for LWD1 in clock regulation via *CCA1* activation. We next examined whether LWD1 activated *CCA1* by binding to the *CCA1* promoter. Results of chromatin immunoprecipitation quantitative PCR (ChIP-qPCR) demonstrated a clear association of LWD1 with the *CCA1* promoter at ZT21, preceding the peak expression of *CCA1* (Fig. 1b), which implies the *bona fide* direct regulation of *CCA1* by LWD1.

**LWD1 interacts with TCP20 and TCP22**. Although LWD1 has no recognizable DNA binding domain, it has 5 WD-repeats that constitute a propeller structure frequently considered a protein–protein interaction interface[19]. Therefore, LWD1 may be recruited to the *CCA1* promoter by interacting with transcription factors that remain to be identified. We used yeast two-hybrid screening to identify LWD1-interacting transcription factor(s) from a prey library of ~1,400 *Arabidopsis* full-length transcription factors[20]. TCP transcription factors were identified as potential interacting proteins of LWD1. An initial screen revealed four class I TCPs (TCP19, −20, −22 and CHE (also known as TCP21)) and TCP3 belonging to class II TCPs (Supplementary Table 1). TCP20 and TCP22 were the predominant interacting partners of LWD1 by their dominant presence (>98%) in the positive clones sequenced (Supplementary Table 1). Both LWD1 and LWD2 could interact with TCP20/TCP22 (Fig. 2a), which is consistent with the overlapping functions of LWD1 and LWD2 (ref. 21). Bimolecular fluorescence complementation assay (BiFC) confirmed that TCP20/TCP22 interacted with full-length LWD1 in *Arabidopsis* nuclei (Fig. 2b) but not truncated LWD1 with deletion of the fourth and fifth WD-repeats and expressed at a comparable level (Fig. 2b; Supplementary Fig. 2). The interactions between TCP20/TCP22 and LWD1 but not truncated LWD1 under the control of their native promoters were further validated in *Arabidopsis* seedlings with luciferase complementation imaging assays (Fig. 2c).

Like other *Arabidopsis* short-period mutants[3,5], the *lwd1 lwd2* mutant has an early-flowering phenotype[21]. The protein–protein interaction between LWDs and TCP20/TCP22 suggested that TCP20/TCP22 might function with LWDs in flowering time control. Indeed, *tcp20-2* (ref. 22), *tcp20-4* (ref. 22) (Supplementary Fig. 3a), and *tcp22-1* (ref. 23) single mutants and the *tcp20 tcp22* double mutant show early transition from the vegetative to reproductive stage under long-day conditions (16-h light/8-h

dark) (Fig. 2d and Supplementary Fig. 3b). The early-flowering phenotype of *tcp* mutants and the interaction between TCP and LWD proteins imply that TCP20/TCP22 also functions in the photoperiodic pathway.

**TCP20 and TCP22 are new clock components**. The expression of *CCA1* is greatly dampened under constant dark condition[21] and can be quickly induced by light signals[24]. To test whether TCP20 and TCP22 are required for *CCA1* activation by light, seedlings of the wild type and *tcp20 tcp22* were first grown under a 12-h light/12-h dark cycle for 10 days and then transferred to constant dark (DD) to minimize the *CCA1* expression. The light-induced and rhythmic expression of *CCA1* was triggered by 1 h of light treatment at DD24. The wild type showed peak *CCA1* expression at DD45 (the following circadian cycle), but the expression amplitude was reduced in *tcp20 tcp22* (Fig. 3a), which indicates that TCP20 and TCP22 are required for the full activation of *CCA1* by light. We next investigated whether the activator role of TCPs on *CCA1* expression results from a circadian equilibrium after a full 24-h rhythm or is a direct regulation by examining the immediate function of TCP20 on the *CCA1* promoter. We fused TCP20 with the ligand-binding domain of glucocorticoid receptor (TCP20-GR) and expressed it under the control of a 35S promoter. With dexamethasone (DEX) treatment, TCP20-GR immediately enhanced the expression of the reporter gene *luciferase* (*LUC2*) driven by the *CCA1* promoter in *Arabidopsis* seedlings (Fig. 3b), which implies a direct activator role of TCP20 on *CCA1* expression.

To examine whether TCP20 and TCP22 regulate the circadian period length of *CCA1*, we introduced *pCCA1::LUC2* into *tcp* single and double mutants by crossing with the reporter line in a wild-type background[14]. Consistent with the transcript levels, the expression of *pCCA1::LUC2* was greatly reduced in *tcp* mutants (Fig. 3c and Supplementary Fig. 3c). Mutations in *TCP20* or *TCP22* shortened the period lengths by 0.7 h as compared with the wild type (Fig. 3c and Supplementary Fig. 3c). Under entrainment condition with a broadened morning peak, the expression of *pCCA1::LUC2* was reduced in the *tcp* mutants as compared with the wild type (Supplementary Fig. 4). This finding is consistent with the short period phenotype, thus an advanced phase of *CCA1* expression in *tcp* mutants. Of note, the short-period and early flowering phenotypes were not further exaggerated in the *tcp20 tcp22* double mutant (Figs 3c and 2d). Therefore, TCP20 and TCP22 function cooperatively rather than redundantly in activating clock genes, possibly because TCP20 and TCP22 can exist as heterodimers[25].

The low activity of the *CCA1* promoter of *tcp20-2* and *tcp22-1* could be partially rescued by expressing *pTCP20::TCP20-Flag* and *pTCP22::TCP22-Flag* in the complementation lines, respectively (Supplementary Fig. 5a). The early-flowering phenotypes could also be complemented (Supplementary Fig. 5b). The degree of complementation was positively correlated with levels of TCP20-Flag or TCP22-Flag proteins in multiple independent lines (Supplementary Fig. 5c).

Like the activity of other clock genes identified previously, that of *TCP20* and *TCP22* promoters oscillated rhythmically under continuous light (LL) in the wild type (Fig. 3d). The transcript levels of *TCP20* and *TCP22* also oscillated slightly under the light/dark cycle (Fig. 3e). Both the promoter activities and transcript levels of *TCP20* and *TCP22* were enhanced in the *cca1 lhy* mutant (Fig. 3d,e), so the expression of *TCP20/TCP22* was likely repressed by CCA1/LHY. Consistent with the repressor role of CCA1 on *TCP20/TCP22*, the peak CCA1 protein level at dawn[4] coincided with the expression troughs of *TCP20* and *TCP22* (Fig. 3d,e). Thus, *TCP20* and *TCP22* are new clock genes under

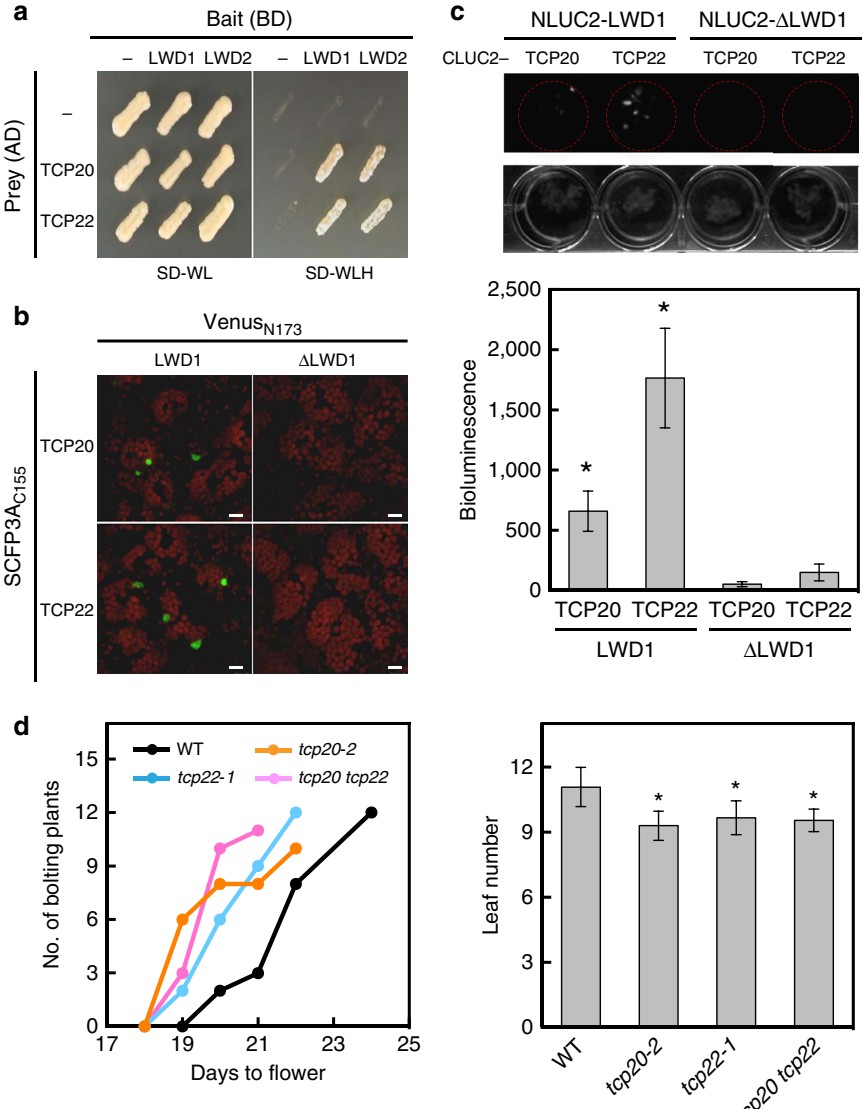

**Figure 2 | TCP20/TCP22 interact with clock proteins LWD1/LWD2 and regulate flowering time.** (**a**) LWD1/LWD2 and TCP20/TCP22 interact on yeast two-hybrid assay. Positive interactions between LWD1/LWD2 and TCP20/TCP22 were shown by the growth on the selection medium without Trp, Leu and His (SD-WLH + 2 mM 3-AT) for the reporter gene *HIS3*. Bimolecular fluorescence complementation (BiFC) assay (**b**) and luciferase complementation imaging assay (**c**) showed that the co-expression between LWD1, but not truncated LWD1 (ΔLWD1), and TCP20/TCP22 reconstituted the activity of fluorescence protein (green signal) and luciferase. For luciferase complementation imaging (LCI) assay, one representative image from three independent is shown. Data are mean ± s.e. (*n* = 3). Asterisks indicate that LCI activity from the interactions of TCP20/TCP22 with LWD1 is significantly higher than that with ΔLWD1 (Student's *t* test; *P < 0.05). (**d**) Early flowering of *tcp20-2*, *tcp22-1*, and *tcp20 tcp22* mutants under long-day conditions. Data are mean ± s.d. (*n* ≥ 10). Asterisks indicate that mutant plants flowered significantly earlier than wild-type plants (Student's *t* test; *P < 0.01). Ten to twelve plants for each genotype were planted for scoring for each biological replicate. Similar results were observed in three independent experiments. Scale bars, 10 μm.

feedback regulation of the morning genes *CCA1/LHY* in the *Arabidopsis* circadian clock.

A canonical CCA1-binding site (CBS) and an evening element (EE) were found in the upstream sequences of *TCP20* ( − 779 to − 772) and *TCP22* ( − 732 to − 725), respectively, relative to the translational start site. We examined whether *TCP20* and *TCP22* are direct targets of CCA1 in the *cca1-1 pCCA1-HA-gCCA1* complementation line. The complementation line accumulated a morning-phased HA-CCA1 protein as endogenous CCA1 (Supplementary Fig. 6a). ChIP-qPCR results showed clear binding of HA-CCA1 to the EE-like region ( − 549 to − 543) in the *CCA1* promoter and EE ( − 728 to − 720) in the *TOC1* promoter *in vivo* as reported[26]. However, HA-CCA1 did not seem to bind to *TCP20* or *TCP22* promoters (Supplementary

Fig. 6b). Similarly, two recent genome-wide surveys of CCA1 target genes did not reveal *TCP20* or *TCP22* (refs 27,28). Thus, CCA1 indirectly represses the expression of *TCP20* and *TCP22*.

**TCP20 and TCP22 directly activates the *CCA1* promoter**. CHE, also known as TCP21, represses *CCA1* via TBS (GGTCCCAC, − 564 to − 557 relative to the translational start site)[12], which is present in a region of the *CCA1* promoter bound by LWD1 (Fig. 1b). We used ChIP-qPCR assays to examine whether TCP20/TCP22 associated with the TBS of the *CCA1* promoter by using anti-Flag or TCP22-specific antisera with a *tcp20-2 pTCP20::TCP20-Flag* complementation line or wild-type plants. Only DNA fragment b containing TBS in the *CCA1* promoter was

enriched in the immunoprecipitated protein–chromatin complex from the complementation line (Fig. 4a) and wild-type plants (Supplementary Fig. 7). These results indicate a direct binding of TCP20/TCP22 to the *CCA1* promoter in *Arabidopsis*.

Electrophoresis mobility shift assay showed that TCP20 and TCP22 could form a protein–DNA complex with a 50-bp

fragment containing TBS ( − 584 to − 535 of the *CCA1* promoter relative to the translational start site) (Fig. 4b). As well, the formation of the TCP–TBS protein–DNA complex could be compromised only in the presence of excess and unlabelled TBS competitor but not the mutated TBS (mTBS) competitor (Fig. 4b), which suggests specific binding of TCP20/TCP22 to

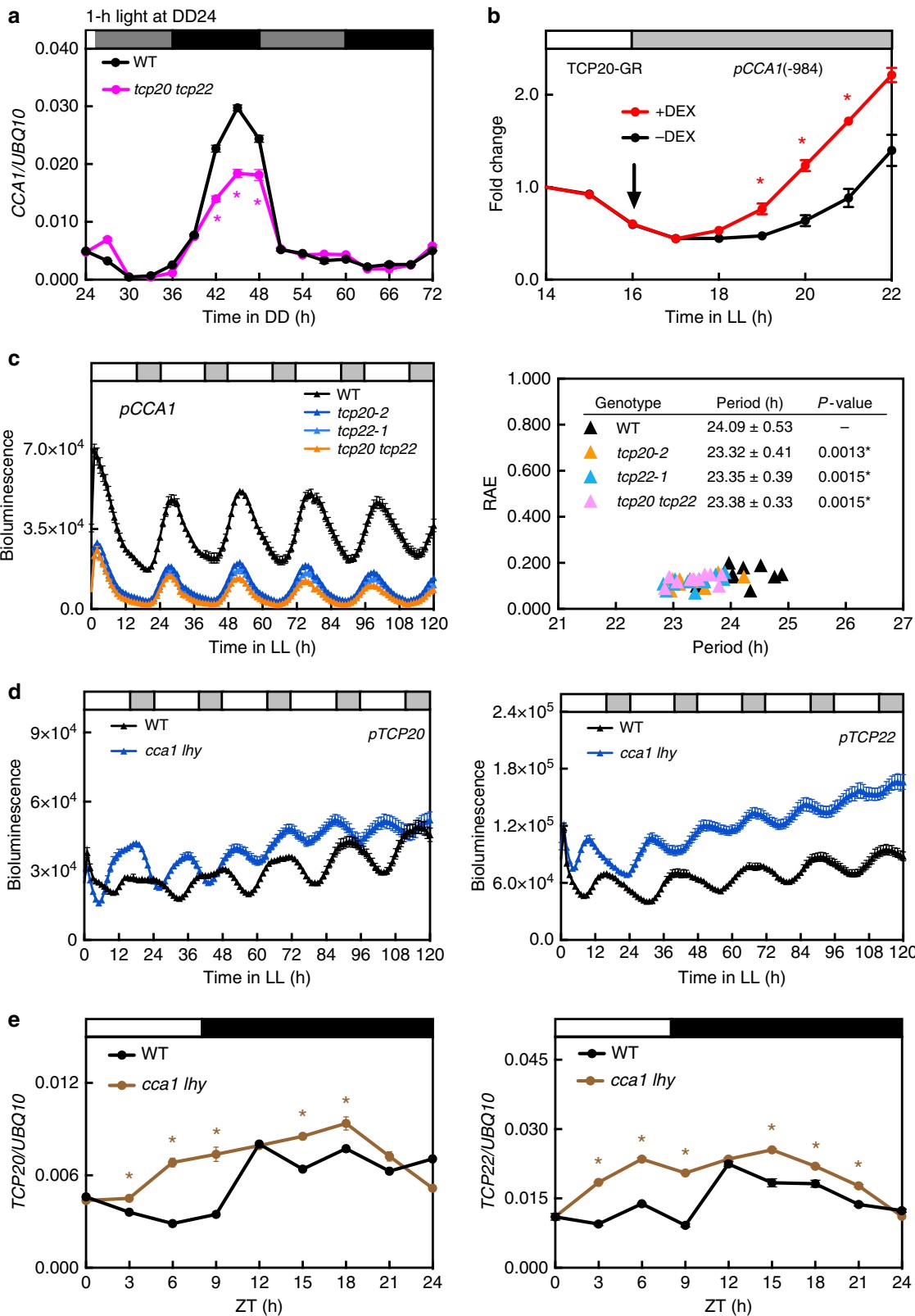

TBS *in vitro*. We further investigated the transcriptional activator role of TCP20 and TCP22 by binding to the TBS in plant cells. To simplify the regulation contributed by TBS, the same 50-bp *CCA1* promoter fragment was used to assay the regulatory role of TCP20 and TCP22 on TBS in the protoplast transient system. The transcriptional activator activity of TCP20 and TCP22 relied on the presence of TBS in this promoter fragment (Fig. 4c). Consistently, the TCP20-dependent activation of the 634-bp *CCA1* promoter was significantly compromised when TBS was mutated (Supplementary Fig. 8a). As expected, the activation was specific for the *CCA1* promoter but not *LHY* promoter (Supplementary Fig. 8b), on which no putative TBS could be identified. Thus, TCP20 and TCP22 activate the expression of *CCA1* via direct and specific binding to the TBS in the *CCA1* promoter. For two functionally redundant genes such as *CCA1* and *LHY* in the circadian clock, their differential regulation by TCPs may increase the flexibility and tunability of the circadian clock.

**TCP20 and TCP22 activate *CCA1* in an LWD1-dependent manner**. Consistent with the activator roles of TCP20 and TCP22 on *CCA1*, *CCA1* promoter activity was significantly greater in multiple independent transgenic lines overexpressing TCP20 (*TCP20ox*) or TCP22 (*TCP22ox*) than in wild-type plants (Fig. 5a and Supplementary Fig. 9a). To address whether interacting with LWDs was crucial for TCP20 and TCP22 to activate *CCA1*, we investigated the functional dependency of TCP20 and TCP22 on LWDs by comparing the *CCA1* promoter activity in TCP20 or TCP22-overexpressing lines in the wild type or in *lwd1 lwd2* backgrounds. Overexpression of TCP20 or TCP22 failed to enhance the *CCA1* promoter activity in *lwd1 lwd2* (Fig. 5b and Supplementary Fig. 9b), even though TCP20 and TCP22 levels were comparable in the wild type and *lwd1 lwd2* (Supplementary Fig. 9c). Thus, TCP20 and TCP22 functioned as activators of *CCA1* only in the presence of LWDs. Consistent with this notion, the trans-activation of TBS-containing promoter by TCP20 was significantly compromised in protoplasts from the *lwd1 lwd2* mutant (Supplementary Fig. 10).

We next examined whether this lack of *CCA1* activation in *lwd1 lwd2* was due to a compromised binding of TCP20 and TCP22 to the TBS in the *CCA1* promoter. TCP20 and TCP22 could still associate with the TBS in the *CCA1* promoter with or without LWDs (Fig. 5c). Our results also showed comparable association of LWD1 with the TBS in the *CCA1* promoter in the wild type and *tcp20 tcp22* double mutant (Fig. 5d), so LWD1 could still be recruited to the *CCA1* promoter in *tcp20 tcp22*, possibly via other transcription factors. In-depth yeast two-hybrid assays revealed additional LWD1- and LWD2-interacting class I TCPs, including TCP7, TCP8, TCP9, TCP14, TCP15 and TCP23 (Supplementary Fig. 11a), which were distributed in various

clades in a phylogenetic tree of class I TCPs (Supplementary Fig. 11b). Among the members in the same clade containing TCP20 and TCP22, TCP14 and TCP23 could also activate *CCA1* expression in a protoplast transient assay (Supplementary Fig. 11c), which suggests that additional class I TCPs have activator roles for *CCA1*. This finding may explain why the expression of *CCA1* was not completely compromised in *tcp20 tcp22* (Fig. 3a,c).

LWD1 protein accumulates throughout the day (Supplementary Fig. 12), thereby rendering accessibility in interacting with both activators (TCP14, TCP20, TCP22 and TCP23) and the repressor (CHE; Supplementary Table 1) of *CCA1* in the TCP superfamily. Future in-depth studies of LWDs are needed to uncover their regulatory roles at different times of the day via different interacting transcription factors and/or target genes.

Collectively, although TCP20/TCP22 could directly target the TBS in the *CCA1* promoter, the binding of TCP20/TCP22 to the *CCA1* promoter was insufficient for *CCA1* activation without the concomitant binding of LWDs. The interaction with LWDs likely confers TCP20/TCP22 with transcriptional activator activity. Therefore, LWDs are the co-activators of TCP20 and TCP22 for the transcriptional activation of *CCA1* (left panel in Fig. 5e).

## Discussion

Transcriptional oscillation can be steadily produced by the combination of three repressors in a repressilator system[29]; however, the amplitude and robustness of biological oscillations could be better maintained when the system contains interlocked positive and negative feedback loops[13]. In the *Arabidopsis* circadian clock, LWD1 and PRR9/PRR7 constitute a positive feedback loop[14]. The expression of the clock co-activator *LWD1* was significantly reduced in *prr9* and *prr7* mutants[14]. Without the full capacity of LWD1 as the co-activator, *CCA1* expression amplitude at dawn could be not maintained, which explains the much-reduced *CCA1* expression peak in the higher-order *CCA1* repressor mutants (*prr9 prr7* and *prr9 prr7 prr5 toc1*)[30].

Previous mathematical modelling of clock gene expression in *Arabidopsis* was mostly based on repressors in the negative regulatory loops[16,31]. The regulation of *CCA1* expression revealed only transcriptional repressors, including PRR9, PRR7, PRR5, TOC1 and CHE. Of note, CHE represses *CCA1* by binding to TBS (ref. 12). We provide direct evidence that the LWD–TCP complex could activate and maintain the robust rhythm of *CCA1* by also binding to TBS. Thus, TCP family members could positively or negatively regulate *CCA1*, and TBS is a target for both negative (CHE) and positive (TCP20 and TCP22) regulators.

Although LNKs function as co-activators of RVEs for the expression of *PRR5* and *TOC1* (ref. 15), they can antagonize RVEs' positive regulation of anthocyanin biosynthesis genes at a

**Figure 3 | A feedback regulation between *TCP20/TCP22* and the morning gene *CCA1/LHY*.** (**a**) *CCA1* transcript level is decreased in the *tcp20 tcp22* mutant. Ten-d-old 12-h light/12-h dark grown seedlings were transferred to DD and illuminated by white light (75 µmol m$^{-2}$ s$^{-1}$) for 1 h at DD24. White, grey and black bars indicate light, subjective light and dark periods, respectively. DD, continuous dark. Data are mean ± s.e. (n = 3 technical replicates). An independent biological replicate showed similar result. Asterisks indicate that *CCA1* transcript levels were significantly reduced in *tcp20 tcp22* (Student's *t* test; *P < 0.005). (**b**) The nuclear transportation of TCP20-GR activated the expression of *pCCA1(−984 to +1)::LUC2*. Seedlings co-infected with *Agrobacteria* carrying 35S-TCP20-GR and *pCCA1::LUC2* were released into constant light (LL) and treated with 40 µM dexamethasone (+DEX) or 0.4% EtOH (−DEX) at LL16 (arrow). Bioluminescent intensity at each time was normalized to that at LL14. Data are mean ± s.d. from three biological replicates. Asterisks indicate that *CCA1* promoter activity was significantly increased by DEX treatment (Student's *t* test; *P < 0.005). (**c**) The expression and period lengths of *pCCA1::LUC2* are reduced in *tcp20-2*, *tcp22-1* and *tcp20 tcp22* mutants. Period length and relative amplitude error (RAE) were calculated by FFT-NLLS analysis according to data from LL48 to LL120 (n ≥ 10 seedlings per genotype). Asterisks indicate that period lengths were significantly reduced in *tcp* mutants (Student's *t* test; *P < 0.01). (**d**) The expression of *pTCP20::LUC2* and *pTCP22::LUC2* is upregulated in the *cca1 lhy* mutant (n ≥ 10 seedlings per genotype). (**e**) *TCP20* and *TCP22* transcript levels are increased in the *cca1 lhy* double mutant under short-day conditions. Data are mean ± s.e. (n = 3 technical replicates). Three independent biological replicates showed similar results. Asterisks indicate that *TCP20* or *TCP22* transcript levels were significantly increased a *cca1 lhy* (Student's *t* test; *P < 0.05).

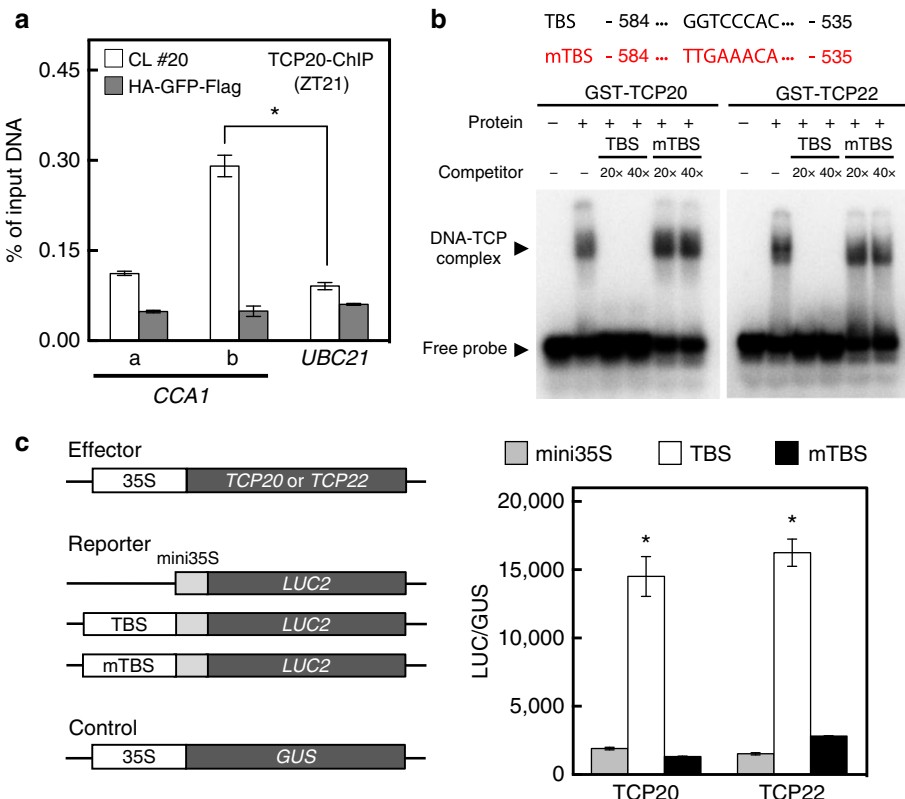

**Figure 4 | TCP20/TCP22 bind to the TCP-binding site (TBS) for *CCA1* activation.** (**a**) TCP20 associates with the TBS-containing region of *CCA1* promoter *in vivo*. ChIP assays involved use of the anti-Flag antisera with *tcp20-2 pTCP20::TCP20-Flag pCCA1::LUC2* complementation line (CL #20) or 35S::HA-GFP-Flag transgenic line. Data are mean ± s.d. (*n* = 3 technical replicates). Asterisk indicates that TCP20 preferentially binds with the amplicon 'b' (Student's *t* test; *P < 0.005). Similar results were observed in two independent experiments. (**b**) EMSA confirmed the specific binding of TCP20 and TCP22 with the TBS. DNA–protein complex was competed with excess (20 × or 40 ×) unlabelled wild-type (TBS) or mutated (mTBS) probes. (**c**) Transient assay in *Arabidopsis* protoplasts of TCP20 and TCP22 possessing activator activity toward promoters containing TBS but not mTBS. Effector and reporter constructs are illustrated. LUC2 reporter activity was normalized to GUS activity from the transfection control *35S::GUS*. Data are mean ± s.d. (*n* = 3). Asterisks indicate that TCP20 or TCP22 preferentially activates the reporter gene driven by promoters containing TBS (mini35S was used as a negative control) (Student's *t* test; *< 0.005). Similar results were observed in three independent experiments.

specific time during the day[32], which indicates a time-dependent switch of roles of LNKs on RVEs toward different target genes. The consecutive expression of the repressors PRR9, PRR7, PRR5, TOC1 and CHE (refs 8–12,16), followed by the activators TCP20/TCP22, during a 24-h cycle likely shapes a precise expression peak of *CCA1* at dawn (right panel in Fig. 5e). Whether the interaction between LWDs and CHE (Supplementary Table 1 and Supplementary Fig. 13) implies a regulatory role of LWDs for CHE remains to be addressed. Additional work is needed to decipher the complex regulation of the *Arabidopsis* circadian clock by TCP transcription factors.

## Methods

**Plant materials and growth conditions.** The wild type and clock mutants *lwd1 lwd2*, *cca1-1* and *lhy-101* in *Arabidopsis thaliana* ecotype Columbia-0 (Col-0) was used for all experiments[21,33,34]. The *cca1 lhy* double mutant was generated by crossing *cca1-1* and *lhy-101* mutants. T-DNA insertion lines *tcp20-2* (SALK_088460c), *tcp20-4* (SALK_041906c) and *tcp22-1* (SALK_027490) were obtained from the Arabidopsis Biological Resources Center. The *tcp20 tcp22* double mutant was generated by crossing *tcp20-2* and *tcp22-1* mutants. For qRT-PCR experiments shown in Fig. 3a, 10-d-old seedlings grown on 0.8% phytoagar solidified half-strength MS (1/2 MS) medium under 12-h light/12-h dark conditions (75 μmol m$^{-2}$ s$^{-1}$) were transferred to constant dark. In Fig. 3e, 18-d-old seedlings were grown under 8-h light/16-h dark conditions (75–100 μmol m$^{-2}$ s$^{-1}$).

**Chromatin immunoprecipitation quantitative PCR assay.** To detect direct target genes of endogenous LWD1 or TCP22, chromatin immunoprecipitation

quantitative PCR (ChIP-qPCR) assays were performed by using 18-d-old wild-type, *lwd1 lwd2* or *tcp20 tcp22* mutants grown under 16-h light/8-h dark on half-strength MS medium. Tissues were harvested at the indicated ZT points. Four hundred plants were treated with 35 ml cross-linking buffer (0.4 M sucrose, 10 mM Tris–HCl, pH 8.0 and 1% formaldehyde) under vacuum for four times, each with 5 min. Glycine was added to a final concentration of 125 mM to quench the excess formaldehyde and to terminate the cross-linking reaction under vacuum for twice, each for 2.5 min, followed by rinsing with 50 ml deionized water for three times. The plants were briefly dried and ground to powder in liquid nitrogen. Five hundred μl powder was then lysed in 0.8 ml cold nuclei lysis buffer (50 mM HEPES pH 7.5, 150 mM NaCl, 0.5% SDS, 0.1% sodium deoxycholate, 1% Triton X-100, 0.1 mM PMSF, 50 μM MG115 and 1 × protease inhibitor cocktail (Roche, Mannheim, Germany)). The lysate was filtered through 100 μm nylon mesh (Calbiochem, La Jolla, CA) in a 2 ml centrifuge tube, and sonicated in a ice bath with the use of Bioruptor (Diagenode, Liege, Belgium) set at high power and 15-s ON/15-s OFF for seven times, each with 5 min. A one-tenth volume of the sonicated lysate was saved for the input fraction control. The LWD1 or TCP22-associated chromatin complexes were immunoprecipitated by incubating the lysate with 3 μl rabbit polyclonal antisera generated with recombinant LWD1 (amino acid residues 1–73) or a polypeptide comprising residues 361–375 (PNQSQASENGGDDKK) of TCP22 for overnight at 4 °C. The immunoprecipitated chromatin complexes were purified by incubating with 50 μl slurry of nProtein A Sepharose (GE, 17-5280-01) pre-equilibrated with 1 mg ml$^{-1}$ salmon sperm DNA and 1 mg ml$^{-1}$ BSA for 3 h at 4 °C.

For HA-tagged CCA1-ChIP, a DNA fragment encoding three tandem repeats of HA-tag was inserted in front of the start codon of a *CCA1* genomic fragment spanning from −1,418 (relative to the translational start site) to the stop codon to generate the *CCA1* complementation construct into *Bam*HI- and *Kpn*I-digested pCAMBIA1390. For Flag–tagged TCP20 or TCP22-ChIP, the overexpressing lines were generated by inserting the coding region of *TCP20* or *TCP22* into *Nco*I and *Bam*HI sites of a modified version of pEarleyGate100 (ref. 35) to generate

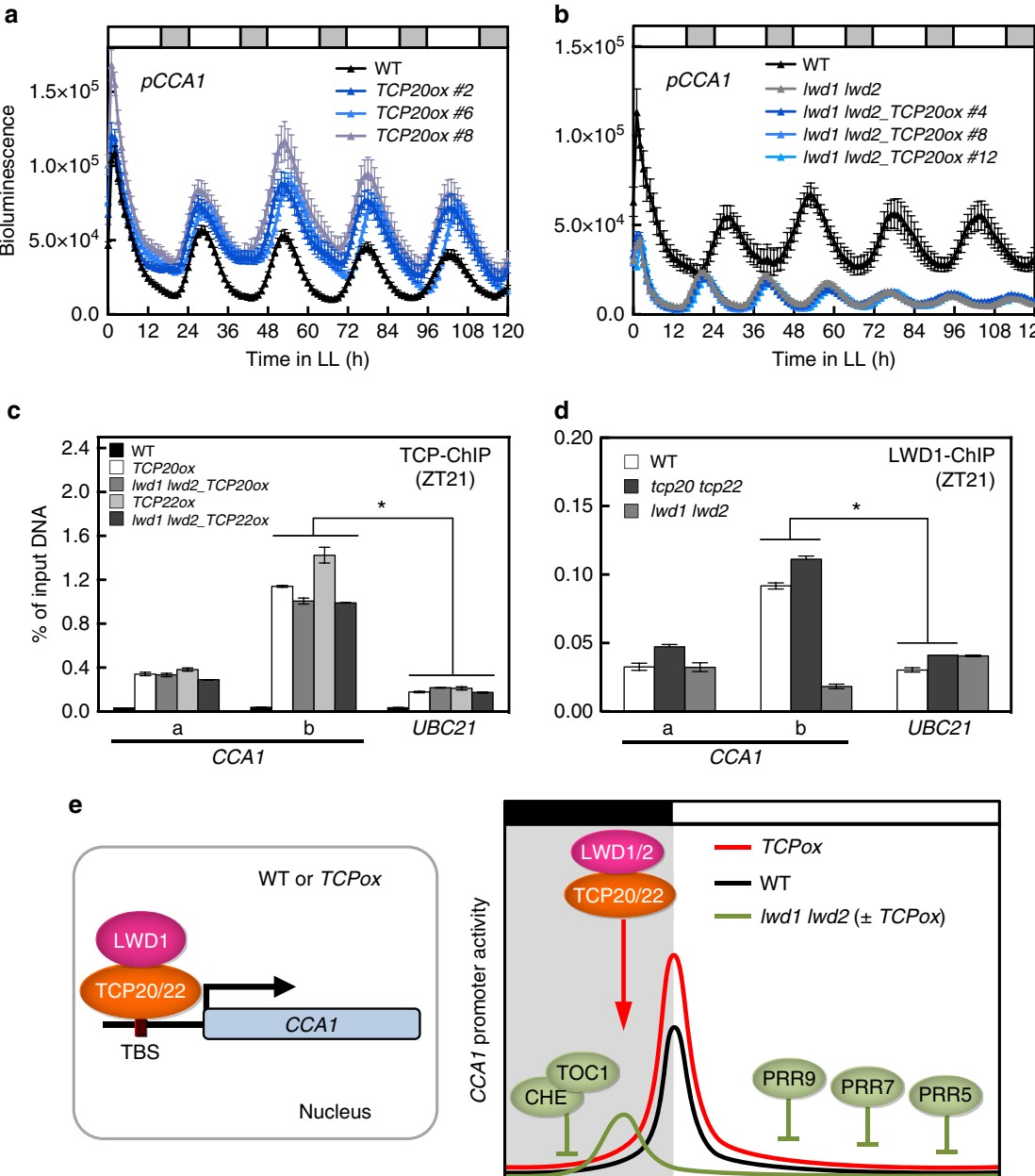

**Figure 5 | LWD1 is the co-activator of TCP20 for the expression of *CCA1* at dawn.** (**a**) *pCCA1::LUC2* expression is increased in TCP20-overexpressing lines. (**b**) Overexpression of TCP20 fails to increase the expression of *CCA1* promoter in *lwd1 lwd2*. Data are mean ± s.e. (*n* = 6–8 seedlings for each independent line; see the corresponding data for *TCP22ox* in Supplementary Fig. 9a,b). Similar results were observed in three independent experiments. (**c**) TCP20/TCP22 associate with the *CCA1* promoter *in vivo* independent of LWDs. Asterisk indicates that overexpressed TCP20/TCP22 in WT or *lwd1 lwd2* backgrounds preferentially binds with amplicon 'b' (Student's *t* test; *$P < 0.005$). (**d**) LWD1 associates with *CCA1* promoter independent of TCP20/TCP22. Data are shown as mean ± s.d. from one representative experiment (*n* = 3 technical replicates). Asterisk indicates that LWD1 preferentially binds to the amplicon 'b' in both WT or *tcp20 tcp22* (Student's *t* test; *$P < 0.001$). Similar results were observed in three independent experiments. (**e**) A diagram of LWD1 and TCP20/TCP22 co-activating the expression of *CCA1*.

CaMV35S (35S) promoter-driven Flag-TCP20 or Flag-TCP22 under wild-type or *lwd1 lwd2* background. ChIP analyses were performed by using 50 μl Sigma monoclonal anti-HA agarose antibody (A2095) or Sigma monoclonal anti-Flag M2 affinity gel (A2225) for HA-CCA1 or Flag-TCP20 and Flag-TCP22.

The affinity beads bound chromatin complexes were washed with 850 μl nuclei lysis buffer for three times, LNDET buffer (0.25 M LiCl, 1% NP40, 1% sodium deoxycholate, 1 mM EDTA, pH8.0) for three times and TE buffer (10 mM Tris–HCl, 1 mM EDTA, pH8.0) for three times. Chromatin complexes were eluted from the beads with use of 200 μl elution buffer (1% SDS, 0.1 M NaHCO₃) followed by the digestion with 2.5 μl Proteinase-K (20 mg ml⁻¹, Invitrogen, Carlsbad, CA) for overnight at 65 °C. DNA released from chromatin complexes was then purified by use of a QIAEXII gel purification kit (QIAGEN, Hilden, Germany) according to the

manufacturer's instruction. Primer sequences used for PCR amplification are listed in Supplementary Table 2. Independent biological replicates of ChIP-qPCR results are shown in Supplementary Fig. 14.

**Yeast two-hybrid analysis.** LWD1-interacting proteins were screened in a prey library of ~1,400 transcription factors by yeast two-hybrid assay[20]. Protein–protein interactions were re-confirmed by using a GAL4-based yeast two-hybrid system (MATCHMAKER GAL4 two-hybrid system, Clontech). Empty pDBLeu (Invitrogen) and pEXP-AD502HA plasmids were used as negative bait and prey controls, respectively. The coding region of *LWD1* or *LWD2* was cloned into the *Xma*I and *Not*I sites of pDBLeu. The pEXP-AD502HA vector is a modified

version of the pEXP-AD502 (Invitrogen) plasmid. Briefly, the T7-HA-MCS derived from pGADT7 (Clontech) was inserted into the SalI and NotI sites of the pEXP-AD502 vector. Coding regions of TCP6, 7, 8, 9, 11, 14, 15, 16, 20, 22 and 23 were cloned into pEXP-AD502HA. SD-WL (-Trp -Leu) plates were used to select for the presence of both bait and prey vectors, and SD-WLH (-Trp -Leu -His) plates with 3-AT (3-amino-1, 2, 4-triazole) were used to select for interactions between the bait and prey proteins. All primers for these constructs for yeast two-hybrid assays are in Supplementary Table 2.

**Bimolecular fluorescence complementation (BiFC) assay.** The *LWD1* full-length coding region or truncated variant lacking amino acids 236–290 (ΔLWD1) was subcloned into hygII-VYNE(R)-pGPTVII, and the coding regions of *TCP20* or *TCP22* were subcloned into kanII-SCYCE(R)-pGPTVII (ref. 36). All primers of these constructs for BiFC are in Supplementary Table 2. Venus$_{N173}$-fused LWD1 or ΔLWD1 was transiently co-expressed with SCFP3A$_{C155}$-fused TCP20, TCP22 or CHE in *Arabidopsis* seedlings by the AGROBEST method[37]. Three days after infection, a GFP filter set was used for detecting the fluorescence signals of SCFP3A$^C$ and Venus$^N$ by Zeiss LSM 780 confocal microscopy.

**Luciferase complementation imaging assay.** Luciferase complementation imaging assays[38] were performed with constructs pLWD1::NLUC2-LWD1, pLWD1::NLUC2-ΔLWD1, pTCP20::CLUC2-TCP20 and pTCP22::CLUC2-TCP22 in *Arabidopsis* seedlings by AGROBEST[37]. Coding regions of *LWD1* (ΔLWD1) and *TCP20/TCP22* were fused with firefly luciferase *NLUC2* (+1 to +1,248, relative to the translational start site) and *CLUC2* (+1,194 to +1,650), respectively, and expressed under the control of their native promoters, *LWD1* (−1,068 to −1), *TCP20* (−2,564 to −1), and *TCP22* (−2,000 to −1).

**Flowering time determination.** *Arabidopsis* seeds were sown on soil and stratified for 3 or 4 days at 4 °C, and then grown under long-day (16-h light/8-h dark) conditions at fluence rate 80–100 µmol m$^{-2}$ s$^{-1}$ at 22 °C. The number of rosette leaves ≥ 5 mm long was recorded for each plant when the primary florescence reached 5 cm above the rosette leaves. This phenotype observation was repeated at least three times.

**RNA preparation and quantitative real-time PCR.** Plants grown under conditions/ZT indicated were harvested and ground in liquid nitrogen. Total RNA was extracted by mixing the tissue powder from 10 plants with 700 µl of 65 °C pre-warmed pine-tree buffer (2% hexadecyltrimethylammonium, 2% polyvinylpyrrolidone K30, 100 mM Tris–HCl pH 8.0, 25 mM EDTA, 2 M NaCl, 0.5 g l$^{-1}$ spermidine, freshly added 2% β-mercaptoethanol)[39]. The extraction mixture was incubated at 65 °C for 5 min. After mixing with 700 µl of chloroform: isoamyl alcohol (24:1), the extraction mixture was centrifuged for 20 min at room temperature. The RNA in the liquid phase was precipitated overnight at 4 °C in a final concentration of 2 M LiCl. RNA pellet was washed by 70% ethanol, briefly air dried, and suspended in DEPC treated mili Q water. Quantitative real-time RT-PCR (qRT-PCR) was performed by using the Applied Biosystem QuantStudio 12K Flex Real-Time PCR System according to the manufacturer's instructions. Sequences and quantities of primers for each gene are listed in Supplementary Table 2. The comparative C$_T$ method was used to determine the relative gene expression, with the expression of *UBQ10* or *UBC21* as the internal control. Mean values of $2^{-\Delta CT}$ ($\Delta C_T = C_{T, \text{gene of interest}} - C_{T, \text{UBQ10 or UBC21}}$) were calculated from three technical repeats. Independent biological replicates of qRT-PCR results are shown in Supplementary Fig. 14.

**Bioluminescence measurement and data analyses.** The coding region of *TCP20* without a stop codon was fused to the ligand-binding domain of glucocorticoid receptor to generate a TCP20-GR effector construct driven by a 35S promoter with modified pEarleyGate100. A 984-bp DNA fragment of *CCA1* promoter (−984 to −1 relative to the translational start site) was cloned and inserted into PstI and NcoI sites to fuse with the *LUC2* gene in the binary vector pCAMBIA1390 (CSIRO, Australia) for constructing the pCCA1(−984)::LUC2 reporter. The effector and reporter constructs were used in the AGROBEST system for transient expression in *Arabidopsis* seedlings under long-day conditions as described[37]. Ten infected seedlings were used for each of three biological replicates for independent induction by 40 µM dexamethasone or 0.4% ethanol mock at the indicated times in bioluminescence analyses.

The pCCA1::LUC2 reporter line for *CCA1* promoter region from −1,418 to −1 relative to the translational start site was used[14]. To generate the pTCP20::LUC2 and pTCP22::LUC2 reporter lines, the upstream 2564- and 2000-bp fragments of the translational start sites for *TCP20* and *TCP22*, respectively, were cloned into the PstI and SalI sites to fuse with the *LUC2* gene in pCAMBIA1390. Seven-d-old seedlings were grown 1/2 MS medium under 16-h light/8-h dark (75 µmol m$^{-2}$ s$^{-1}$), then transferred to continuous light (30–35 µmol m$^{-2}$ s$^{-1}$) at ZT0. Seedlings were imaged every 1 h for 5 d. Bioluminescence activity was measured[14]. Period length was calculated by using the Biological Rhythms Analysis Software System (BRASS[40], available at http://www.amillar.org) and analysed with use of fast Fourier transform-non linear least squares (FFT-NLLS).

**Electrophoretic mobility shift assay.** The coding region of *TCP20* or *TCP22* was cloned into the pET42a (+) vector (Novagen) to generate GST-tagged TCP20 or TCP22. Recombinant GST-TCP20 and GST-TCP22 were expressed in *Escherichia coli* BL21(DE3) and purified by using glutathione-affinity agarose beads (GE) following the manufacturer's protocol. To generate radioactive probes used in electrophoretic mobility shift assay (EMSA), two oligonucleotides for the wild-type or mutated TBS sequence with 10-nt complementary 3′-ends were annealed at a final concentration of 50 µM for each oligonucleotide in TE buffer. The partially annealed DNA were filled-in and radiolabeled for 30 min at 37 °C in a 10 µl reaction mixture comprised 25 µCi of (α-$^{32}$P)dCTP (specificity equals to 3,000 Ci per mmol; Perkin-Elmer), 2.5 unit Klenow fragment of DNA polymerase I (New England BioLabs), labelling buffer (200 mM HEPES, 50 mM Tris–HCl pH8.0, 5 mM MgCl$_2$, 10 mM β-mercaptoethanol, 20 µM for each of dATP, dTTP and dGTP). The oligonucleotides used to generate radiolabeled DNA probes and unlabelled competitors are listed in Supplementary Table 2. EMSA was performed[41] by mixing 0.05 pmol of the probe with purified recombinant protein in the binding buffer (35 mM KCl, 15 mM HEPES 7.5, 0.4 mM EDTA, 6% glycerol, 50 ng µl$^{-1}$ poly(dI-dC)) with or without competitors at room temperature for 10 min. DNA–protein complexes were separated by a 4.5% Tris/Borate/EDTA polyacrylamide gel under 0.5 × Tris-Glycine buffered electrophoresis system, and fixed in 10% acetic acid for autoradiography detection.

**Transient expression in *Arabidopsis* protoplasts.** DNA fragments relative to the translational start site of *CCA1* including 50-bp (−584 to −535) fused with 35S minimum promoter and 634-bp (−634 to −1) of the *CCA1* promoter containing the wild-type or mutated TCP binding site (TBS or mTBS, respectively), and *LHY* promoter (−1,661 to −1) (ref. 14) were ligated to the upstream of *LUC2* in the LUC2-modified pJD301[42] vector to generate reporter constructs in protoplasts. The coding regions of *GFP*, *LWD1*, *TCP14*, *TCP20*, *TCP22* or *TCP23* were tagged with epitope HA and sub-cloned into the p326-GFP (ref. 43) vector for transient overexpression under the control of the 35S promoter. For transient assay in *Arabidopsis* protoplasts, 150 leaves of 3-wk-old plants grown under 12-h light/12-h dark condition were to isolate mesophyll protoplasts by the sandwich method. Briefly, the leaves were digested in the enzyme solution (1% cellulase R10, 0.25% macerozyme R10, 0.4 M mannitol, 20 mM KCl, 20 mM MES, pH5.7, 10 mM CaCl$_2$, 5 mM β-mercaptoethanol, 0.1% BSA) to obtain 2.5 × 10$^6$ mesophyll protoplasts for transfections[44,45]. An amount of 2.5 × 10$^5$ protoplasts was co-transfected with 2 µg reporter construct, 5 µg 35S::TCP(s)-HA as the effector and 10 µg pBI221 (35S::GUS) as the transfection control. Transfected cells were incubated at 22 °C overnight and harvested for LUC2 and GUS reporter assays.

**Phylogenic analysis.** The phylogeny analysis of full-length protein sequences class I TCP members was performed with the one-click mode of the web-based phylogeny tool Phylogeny.fr (http://www.phylogeny.fr/)[46]. Sequence data for genes referred to in this study can be found in the Arabidopsis Genome Initiative data library with the following locus identifiers: *LWD1 (At1g12910)*, *LWD2 (At3g26640)*, *CCA1 (At2g46830)*, *LHY (At1g01060)*, *PRR7 (At5g02810)*, *PRR9 (At2g46790)*, *TOC1 (At5g61380)*, *TCP3 (At1g53230)*, *TCP6 (At5g41030)*, *TCP7 (At5g23280)*, *TCP8 (At1g58100)*, *TCP9 (At2g45680)*, *TCP11 (At2g37000)*, *TCP14 (At3g47620)*, *TCP15 (At1g69690)*, *TCP16 (At3g45150)*, *TCP19 (At5g51910)*, *TCP20 (At3g27010)*, *TCP21/CHE (At5g08330)*, *TCP22 (At1g72010)*, *TCP23 (At1g35560)*, *UBQ10 (At4g05320)* and *UBC21 (At5g25760)*.

**Data availability.** The authors declare that all data supporting the findings of this study are available within the article and its Supplementary Information Files or are available from the corresponding author upon request.

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

## Acknowledgements

We thank Jen Sheen and the Wu lab members for helpful discussion. This research was supported by an Academia Sinica postdoctoral fellowship to H.-L.T., research grants from the Ministry of Science and Technology, Taiwan, to S.-H.W. (100-2311-B-001-028-MY3 and 103-2311-B-001-009-MY3), C.-P.H. (104-2627-M-001-003- and 103-2627-M-001-004-), grants from the Next Generation Biogreen 21 Program (SSAC grant PJ01107102) and the Basic Science Research Program funded by the Ministry of Education (NRF- 2013R1A1A2010131) to J.C.H.

## Author contributions

J.-F.W., H.-L.T., I.J., C.-P.H. and S.-H.W. designed the research, analysed the data and wrote the article. J.-F.W., H.-L.T., I.J., Y.-C.W., Y.-H.L. and C.-W.C. performed the research. J.-W.C. and C.-P.H. supervised the computational work. Y.W. and J.C.H. contributed experimental materials.

## Additional information

**Competing financial interests:** The authors declare no competing financial interests.

