## [Peer Review File · Nature Communications]

Reviewers' comments:

Reviewer #1 (Remarks to the Author):

This study used mathematical modeling to examine the robustness of clock performance when LWD1 activates only PRR9 (model 1) or both PRR9 and CCA1 (model 2). As predicted by the modeling approach, positive regulation of CCA1 by LWD1 was experimentally demonstrated. Moreover, authors identified two new clock proteins that interact with LWD1 to activate CCA1. The approach they used is interesting and finding new clock components is exciting. However whether these newly found components and its regulatory pathways play an important role for clock function is not clear. Their finding could be just an addition of new complexity to the system that is already complex. Authors need to demonstrate what roles these new components play more clearly. Followings are my major concerns.

1. By running the model with randomly chosen parameter sets, they showed that likelihood that model 2 generates proper oscillatory dynamics was 40 times more than that of model 1. I think this comparison should be done using null model that considers no effect from LWD1. Adding the effect of LWD1 should improve the performance of null model if authors intend to show the importance of positive feedback to enhance robustness of clock function. Authors should check this first.

2. They examined the Pokhilko model as the basis to incorporate LWD1 effect. The number of parameters is much larger than the simplified model they used before. How parameter values are chosen? Did they perform similar analysis as random choice of parameter sets? They concluded that the integrity of the Pokhilko model was maintained even after incorporating LWD1 as an activator of PRR9 and CCA1. But if this occurs only limited number of parameter set, their conclusion would be misleading.

3. Mutants tcp20/tcp22 showed early flowering phenotype, but it is not so significant because the differences between wt and mutants were only several days. I am not sure what roles these new clock components play.

4. Their experiments are performed mostly in continuous light condition. Even if peak timing of CCA1 is different between wt and mutants tcp20/tcp22, light signal may reset the clock and this difference might be less pronounced L/D cycle. If authors emphasize the role of TCP to set the phase of morning clock genes, they need to show how mutants behave under L/D conditions.

Reviewer #2 (Remarks to the Author):

Review for Wu et al.
NCOMMS-16-08634A-Z

In this manuscript, the authors build on their earlier identification of a positive feedback loop involving LWD1, LWD2 and PRR9 in the Arabidopsis circadian clock (Wu et al 2008 and Wang et al 2011). Mathematical modelling and ChIP suggests that LWD1 directly regulates CCA1 expression although it lacks a DNA binding domain. Yeast two-hybrid then identifies interaction between LWD1 and TCP20/TCP22, and a highly convincing set of experiments ensues to gradually show that TCP20 and TCP22 activate CCA1 expression in the presence of LWD1, while inhibiting CCA1 expression in the absence of LWD1. This is a commendable amount of work, and the experiments are beautifully

crafted.

Although the significant amount of novel information described here leads to original conclusions that will be of clear interest to plant circadian clock researchers, I am not certain whether the novelty of yet another feedback loop in the Arabidopsis clock is evident to scientists outside this specific niche. Personally, I greatly enjoyed reading your manuscript, and I think it would benefit from addressing the following issues.

-Statistical tests are missing to show the significance of the results in Fig. 1b, 2c, 3a,b,e, 4a,c, and 5c,d.

-In addition to TCP20 and 22 (Figure 2), have the authors considered looking at in vivo interaction between LWD1 and TCP19 and 21 (CHE)? This is particularly relevant for CHE as it has been shown to bind the same site as where LWD1 binds. This potential interaction is alluded to later in the manuscript and could explain why LWD1 still binds the CCA1 promoter in the absence of TCP20/22 (Fig. 5d). However, it is very unclear to me why a second yeast two-hybrid with LWD1 is needed in addition to the one already described in Figure 2? Does it really make sense to separate these into two experiments in two places of the manuscript? In my opinion it is stronger to put all of that together in either place of the manuscript.

-P6L160; the statement 'TCP20 and TCP22 are required for activating CCA1' seems a long stretch, considered CCA1 still gets activated in the tcp20/tcp22 double mutant, up to ~70% of wt levels. Certainly that means other factors are involved in 'waking up' CCA1 independent of TCP20/22.

-The results show that TCP20 and TCP22 are both required to activate CCA1 (i.e. the double mutant has identical effects as each single mutant). How does that match with the results in Figure 3b where only TCP20 is induced? How are the effects of overexpression of only one of them explained (figure 5a-b)?

-Modelling using both a new simple model as well as an integrated complex model suggests that it is feasible that LWD1 positively regulates both CCA1 and its repressor PRR9. The rest of the manuscript suggests that TCP20/TCP22 can positively and negatively regulate CCA1 expression depending on the presence of LWD1. Given the complexity, a mathematical model would really help at the end of the paper to assess whether the full set of results, including dynamics of LWD1/TCP20/TCP22, actually explains clock behaviour. This bit of modelling could complement the visual model in Figure 5e.

Minor points:

-The modelling in Figure 1 only involves LWD1; are dynamics affected by including LWD2? If not, should all labels possibly read 'LWD1/2'?

-P2L29, first sentence of abstract states that 'clock is driven' by CCA1/LHY and TOC1. This statement lacks nuance, when knock-outs of these genes are still rhythmic.

-P3L70, sentence starting 'Here, we...' needs rewriting to improve clarity.

-P4L83, title needs rewriting: LWD1 does not use mathematical modelling to activate PRR9 and CCA1.

-P4L94, statement about two parameters to search for needs clarification.

-P5L142, the data in Figure 2c would benefit from a little more explanation than one single sentence.

-P6L160, sentence starting 'We next...' needs rewriting to improve clarity.

-P7, section on 'feedback loop'; I always assumed interactions within the full feedback system are called a 'loop' when the interactions both ways are direct? Data clearly show CCA1 does not directly bind the TCP20/22 promoters, therefore there is no loop between CCA1 and TCP20/22?

-CCA1 and LHY are partially redundant. Could you briefly discuss on the relevance of a repressor/activator complex that only acts on one of the two?

Reviewer #3 (Remarks to the Author):

In this paper by Wu et al, entitled "LWD-TCP complex wakes up the morning gene CCA1 in

Arabidopsis," the authors define new interactions and roles for the LWD proteins and the TCP transcription factors. *lwd1/2* mutants have a major role maintaining the amplitude and period of the circadian clock. The core-clock transcription factor CHE, which is also a TCP transcription factor, negatively regulates the amplitude of the CCA1. The authors have identified a novel, direct interaction between TCP20 and TCP22, and with LWD1 or LWD2. Through modifying and extending an ODE-based mathematical modeling of the clock (Pokhilko et al. 2012), the authors predict a co-activator role for LWD1 at the CCA1 promoter, and suggest that the LWDs are recruited to the promoters through an interaction with TCP20 and 22. TCP20 and 22 can directly bind to the CCA1 promoter in an LWD1/2 independent manner. However, LWD1 recruitment the CCA1 promoter does not require TCP20 or 22, which is likely due to redundancy in the type 1 TCP family, which broadly interact with LWDs. These findings identify the first transcriptional activators identified for CCA1. The paper is well written, data is clear and presented consistently. However, I have the following concerns:

Major concerns:

1. The authors show indirectly that LWD1, LWD2, TCP20 and TCP22 are necessary for activation by through the use of bioluminescence reporters in vivo, yet the authors do not show a complementation experiment where they observe rescue of the loss of amplitude of the CCA1 reporter. The authors should show that the loss of amplitude of the reporter is due to absence of the gene by complementing with a transgene for TCP20 or TCP22 into *tcp20* or *tcp22*, respectively. The reviewer is concerned that the loss of amplitude is due to co-suppression of the CCA1::LUC reporter by TDNA insertions in the mutant backgrounds, and not necessarily reflective of amplitude of CCA1 (Gao Y, Zhao Y. Epigenetic Suppression of T-DNA Insertion Mutants in Arabidopsis . *Molecular Plant*. 2013;6(2):539-545. doi:10.1093/mp/sss093). If the effect on amplitude of the reporter is direct, then amplitude of the *tcp 20* CCA1:LUC line should be rescued by complementing with a TCP20 transgene.
2. The authors suggest that concomitant recruitment of LWD1 to promoters with TCP will lead to activation. In figure 4c, TCP20 and TCP22 can induce the expression of a TBS reporter in a transient transfection assay. If LWD is important for co-activation by TCPs, then co-transfection of LWD1 in similar experiments should lead to increased expression of the reporter. Alternatively, if endogenous LWD is a concern, then the authors could show that transfection of effectors and the reporter into *lwd1/lwd2* protoplasts leads to lower transactivation compared to wild type. This would show directly in a transient assay that LWDs can act as co-activators in concert with the TCPs through the TBS.

Minor concerns:

1. In figure 5e, the model, the authors suggest that LWD1 and TCP20/22 are recruited to the CCA1 promoter directly to activate transcription, and without LWD1 and LWD2 TCPs cannot induce CCA1, and as drawn, TCP20/22 directly recruits LWD1. However, LWD1 can be ChIPed at the promoter in the absence of TCP20/22 similarly as with (or better maybe). Likewise, CCA1 is still expressed in the *lwd lwd2* background, albeit with reduced amplitude and period, so the sections on the left are a little misleading. The model should better reflect the data.
2. In figure 4c does "Vector" in the graph refer to mini35S in the right schematic or a mock control where the TBS reporter is transformed without co-expressing TCP20? A mock control is required to determine if the TBS vector is strongly induced without addition of the TCP effector in protoplasts. A TBS alone control without a TCP effector would be an appropriate control for this experiment, and the experiment performed in supplemental figure 8.
3. Table 2, supplemental data: typo- Dimmension should be spelled Dimension

Reviewer #4 (Remarks to the Author):

The authors will recognize some of these comments as I reviewed an earlier version at another journal. The manuscript is improved since I last saw it. Some of my major concerns from that previous submission have been addressed and I think the manuscript provides good evidence for their major

arguments that the LWD transcriptional regulators regulate the circadian oscillator gene CCA1 through the TCP class of transcription factor. There are still some areas of concern that remain. If those are unchanged from the previous version I have used the same text as the last time I saw this work.

The authors address an area of interest by investigating the the activators of LHY/CCA1 expression at dawn. The authors use a mix of modeling, yeast 2 hybrid, mutant studies, transient activation and CHIP to test the roles of LWD1/2.

I have most concerns regarding the modeling. The results section starts with a paragraph describing a modeling approach which the authors describe as comparing the potential for LWD1 to regulate PRR9 alone with models in which LWD1 regulates both PRR9 and CCA1 together. This is a misleading description because in the model CCA1 and LHY, and PRR7 and PRR9 are collapsed each in to single genes, and therefore the model compares the effect of LWD1 regulating both PRR9/PRR7 or with LWD1 regulation all of PRR9, PRR7, CCA1 and LHY. Based on the authors own data it would seem that particularly in the context of LWD1 this simplification of the model is inappropriate and makes the modeling predictions unhelpful in the context of the work presented in this manuscript. The authors have now added a re-analysis of the Pokhilko 2012 model which is a much better tool for their investigation. The remodeling in the Pokhilko model described in the second paragraph supports the conclusion that LWD1 regulates PRR9 and not PRR7.

A CHIPseq identified an interaction between LWD1 and the CCA1 promoter at ZT21, around the time when transcriptional activation of CCA1 begins. Because LWD1 has no DNA binding motif the authors looked for potential interacting proteins using yeast two hybrid and identified four potential interactors of the class I TCP transcriptional regulators. One of these, TCP21/CHE has been shown previously to repress CCA1 expression. A class II TCP3 was also found as a potential interactor. The dominant clones sequenced were TCP20 and 22, which prompted the authors to investigate these further. LWD2 was also shown to interact with TCP20/22. BiFC was used to confirm the interaction with LWD1. The TCP20, TCP22 single and double mutants had short period circadian rhythms and a short day flowering phenotype.

The expression levels of CCA1 were found to be lower in TCP20 TCP22 double mutants. The authors used transient induction experiments to demonstrate that TCP20 is an activator of CCA1. The tcp 20 and tcp 22 mutants had a slight short period phenotype.

The promoters of TCP20 and 22 oscillated under continuous light and under light and dark cycles. The authors provide evidence that the expression of TCP20 and TCP22 are under circadian control. The authors show that CCA1 and LHY do not directly repress TCP20 and TCP22.

The TCP binding site (TBS) is present only in the promoter of CCA1 and not LHY and the authors found that the TBS was bound by TCP20 and TCP22. The authors provide good evidence that TCP20 and TCP22 bind and regulate the CCA1 and not the LHY promoter

I find the conclusion that LWD1 and LWD2 are required for the induction compelling based on the TCPox studies in the lwd1 lwd2 mutant background. Perhaps surprisingly, TCP20 and TCP22 bound the CCA1 promoter in lwd1 lwd2 null backgrounds which lead the authors to conclude that LWD are transcriptional co-activators of TCP20 and TCP22, which I think is reasonable interpretation of the data.

Major Comments

As I have mentioned above. I am was very concerned by the modelling. approach when I received this

manuscript the first time. That modeling approach remains and is described in the first paragraph of the results but it is now supplemented by a second approach which is described in lines 99 onwards.

I wrote about the modelling. approach that is described in the first paragraph " The approach was to use a reduced model form considering LHY/CCA1 as a single gene and PRR9/PRR7 also as a single gene. The goal was to understand the function of LWD1. Since LWD1 binds only CCA1 and not LHY, and LWD1 regulates PRR9 but apparently not PRR7, it seems to me that this is a simplification too far. The experimental data demonstrate that the assumption that CCA1/LHY and PRR7/PRR9 can be considered as functional equivalents is false. For some studies the assumption, whilst false, might not produce misleading results, however in this context the goal is to understand the potential functions of LWD1, which regulates only one of each of these two gene pairs, it might be that the simplification of the model must inevitably obscure the likely roles of LWD1. Therefore almost any predictions from the model concerning LWD1 have a high probability of being false. I can not see any justification for treating these genes pairs as single genes when it is known that they are targeted differently by LWD1 and it was this regulation that the model was specifically designed to investigate. The explanation of the model is far from clear "among 2.4x10⁸ random parameter sets examined, 1,004 sets met our criteria (see Methods) for Model II as compared with only 27 of 3x10⁸ random parameter sets for Model I." What criteria for what? And "Thus, faithful oscillation was greater than 40 times easier to obtain when LWD1 also activates CCA1, despite having two more parameters to search for in Model II. " What is meant by a faithful oscillation, how can it be 40x easier to obtain? This all needs much better explanation. All my criticisms concerning this aspect of the manuscript remain.

The authors have improved on the original version by adding a new part to the modelling. in which they examine the role of LWD1 in the Pokhilko model (starting at line 99 of the MS). This is a more sensible strategy than the modelling. described in the first paragraph of the MS. The manuscript would be improved by removal of the first modeling strategy. I do not think it provides meaningful insight, whereas the use of the Pokhilko model is useful.

The authors still do not explain why some of the experiments were performed in constant dark (DD). To interpret the data it needs to be explained why the experiments were performed in constant dark.

The authors now report that TCP20 and 22 mutants have a very small phenotype of slightly shorter than wild type

In my review of a previous version of the manuscript I made negative comments about the figure legends. These comments still apply.

Figure 1 the legend is unclear. How can there be period estimate differences with errors for the deterministic models? Surely deterministic models will always produce the same period estimate? If these were the means of the different simulations, how is this informative when a huge range of parameters were used? More explanation is needed. Why were amplicons a and b chosen, what is their significance?

Figure 2 the legend is too incomplete to understand the data. Specifically what is being shown? The authors could expand the figure legend and still comply with the article length restrictions by reducing the length of the final conclusion which is a bit repetitive of the text in the rest of the manuscript.

Minor Comments

I criticized the abstract in my review of a previous version of the paper. Those criticisms still apply. The abstract contains many inaccuracies and over simplifications of the circadian system which are misleading. Such as the following which I believe to be incorrect "CCA1 initiates and sets the circadian phase, which requires its peak expression at dawn." The phase is a result of many circadian components and CCA1 is usually considered to peak shortly after dawn. And the Arabidopsis circadian clock is driven by a double negative feedback loop formed by the morning genes CCA1/LHY and the evening gene TOC1". I disagree, these are only part of the system.

Diurnal is used incorrectly in this manuscript. Diurnal is the antonym of nocturnal and should not be used to describe light and dark cycles.

The method still do not state what media the plants were grown on. This is essential information

It was not clear to me if LWD2 interacted with the same members of the TCP clade as LWD1. Is this described anywhere? Apologies if I missed those data. It would be good to make clear whether those interactions were tested, and if they were, what the result was. If they were not tested I think that is OK, but this needs to be clear.

Responses to the Reviewers comments

Reviewer #1:

This study used mathematical modeling to examine the robustness of clock performance when LWD1 activates only PRR9 (model 1) or both PRR9 and CCA1 (model 2). As predicted by the modeling approach, positive regulation of CCA1 by LWD1 was experimentally demonstrated. Moreover, authors identified two new clock proteins that interact with LWD to activate CCA1. The approach they used is interesting and finding new clock components is exciting. However whether these newly found components and its regulatory pathways play an important role for clock function is not clear. Their finding could be just an addition of new complexity to the system that is already complex. Authors need to demonstrate what roles these new components play more clearly. Followings are my major concerns.

1. By running the model with randomly chosen parameter sets, they showed that likelihood that model 2 generates proper oscillatory dynamics was 40 times more than that of model 1. I think this comparison should be done using null model that considers no effect from LWD1. Adding the effect of LWD1 should improve the performance of null model if authors intend to show the importance of positive feedback to enhance robustness of clock function. Authors should check this first.

Response:

Perhaps our description misled the reviewer to think our mathematical modeling was meant to emphasize the positive feedback of LWD1 and PRR9 in the clock. In fact, in this study, we intended to explore additional regulatory roles of LWD1 and found that the clock is more sustainable only when LWD1 regulated both *PRR9* and *CCA1*.

In this report, we showed that LWD1 protein does not oscillate significantly in a 24-h cycle (Supplementary Fig. 12). Thus, LWD1 may not provide a direct oscillating cue for the clock genes. This may explain why without including LWD1 in the modeling, proper oscillation could still be obtained in many previous modeling works (Mol. Syst. Biol., 2012, 8: 574; Mol. Syst. Biol., 2013, 9: 650; Mol. Syst. Biol., 2015, 11: 776). These models will resemble the null models suggested by the reviewer. It is possible that the effects of LWD1 are indirectly included in their parameters and model settings. However, the previously developed Pokhilko model (“null model”) does not have the ability to describe the clock defects observed in the *lwd1 lwd2* mutant. By integrating LWD1 into the Pokhilko model while maintaining most of the desirable properties of this model, we have indeed compared models with and without LWD1, the latter being the original Pokhilko model (Supplementary Fig. 1).

2. They examined the Pokhilko model as the basis to incorporate LWD1 effect. The number of parameters is much larger than the simplified model they used before. How parameter values are chosen? Did they perform similar analysis as random choice of parameter sets? They concluded that the integrity of the Pokhilko model was

maintained even after incorporating LWD1 as an activator of *PRR9* and *CCA1*. But if this occurs only limited number of parameter set, their conclusion would be misleading.

Response:

In this study, we first used a simplified/targeting modeling to reveal positive regulatory roles of LWD1 in the expression of both *PRR9* and *CCA1*. We then confirmed that the dual roles of LWD1 could be seamlessly integrated into a more complex model (the Pokhilko model). The choices of parameters are described in a Supplementary Note. Briefly, most of the parameter values were kept the same as the original settings in the Pokhilko model. We only removed three parameters from the Pokhilko model (n_1 , n_4 , and n_7) and added 3 parameters for single activation of *PRR9* by LWD1 or 6 parameters for double activation of both *PRR9* and *CCA1* by LWD1 as shown in Table 2 in the Supplementary Note. Those additional parameters were chosen randomly with some minor adjustments. We concluded that the modified Pokhilko model would work fine in both wild-type and mutant conditions, meaning that it is possible to incorporate LWD1 in the Pokhilko model. All models need to work with certain limitations in their parameters (or parameter ranges). In the case of Pokhilko model, the dynamics of the system is not much changed with the fixed parameter values in the original model.

3. Mutants *tcp20/tcp22* showed early flowering phenotype, but it is not so significant because the differences between wt and mutants were only several days. I am not sure what roles these new clock components play.

Response:

In addition to TCP20 and TCP22, additional class I TCP family members can interact with LWD1 and have trans-activating activities toward *CCA1* as shown in Supplementary Fig. 11. Functional redundancy among class I TCPs may explain why the *tcp20 tcp22* double mutant does not have a more obvious early-flowering phenotype as in *lwd1 lwd2* mutant. However, our additional data clearly showed the TCP20 and TCP22 are new clock components and activators of the morning gene *CCA1*.

4. Their experiments are performed mostly in continuous light condition. Even if peak timing of *CCA1* is different between wt and mutants *tcp20/tcp22*, light signal may reset the clock and this difference might be less pronounced L/D cycle. If authors emphasize the role of TCP to set the phase of morning clock genes, they need to show how mutants behave under L/D conditions.

Response:

As suggested by the reviewer, we performed bioluminescence assays to examine the expression of *pCCA1::LUC2* in *tcp20-2*, *tcp22-1* and *tcp20 tcp22* mutants under long-day conditions. *CCA1* promoter activity remained reduced in *tcp* mutants (Supplementary Fig. 4a), similar to that under constant light (Fig. 3c). As pointed out by the reviewer, light indeed resets the clock each day. However, a phase-advanced expression of *CCA1* can still be observed in the *tcp* mutants (Supplementary Fig. 4b).

Reviewer #2:

Review for Wu et al.
NCOMMS-16-08634A-Z

In this manuscript, the authors build on their earlier identification of a positive feedback loop involving LWD1, LWD2 and PRR9 in the Arabidopsis circadian clock (Wu et al 2008 and Wang et al 2011). Mathematical modelling and ChIP suggests that LWD1 directly regulates CCA1 expression although it lacks a DNA binding domain. Yeast two-hybrid then identifies interaction between LWD1 and TCP20/TCP22, and a highly convincing set of experiments ensues to gradually show that TCP20 and TCP22 activate CCA1 expression in the presence of LWD1, while inhibiting CCA1 expression in the absence of LWD1. This is a commendable amount of work, and the experiments are beautifully crafted.

Although the significant amount of novel information described here leads to original conclusions that will be of clear interest to plant circadian clock researchers, I am not certain whether the novelty of yet another feedback loop in the Arabidopsis clock is evident to scientists outside this specific niche. Personally, I greatly enjoyed reading your manuscript, and I think it would benefit from addressing the following issues.

-Statistical tests are missing to show the significance of the results in Fig. 1b, 2c, 3a,b,e, 4a,c, and 5c,d.

Response:

We have added statistical test results to all figures listed by the reviewer and also those newly added in the revised manuscript.

-In addition to TCP20 and 22 (Figure 2), have the authors considered looking at in vivo interaction between LWD1 and TCP19 and 21 (CHE)? This is particularly relevant for CHE as it has been shown to bind the same site as where LWD1 binds. This potential interaction is alluded to later in the manuscript and could explain why LWD1 still binds the CCA1 promoter in the absence of TCP20/22 (Fig. 5d).

Response:

As suggested, we have performed BiFC assays to test the interaction between CHE, the repressor of *CCA1*, and LWD1 (Supplementary Fig. 13). The results of BiFC showed a positive interaction between CHE and LWD1 in plant cells, thereby supporting that LWD1 interacts with multiple members of class I TCP members, including both activators and a repressor of *CCA1*.

However, it is very unclear to me why a second yeast two-hybrid with LWD1 is needed in addition to the one already described in Figure 2? Does it really make sense to separate these into two experiments in two places of the manuscript? In my opinion it is stronger to put all of that together in either place of the manuscript.

Response:

The yeast two-hybrid shown in Fig. 2a was to confirm the results from the screening of transcription factor library for LWD1-interacting proteins, with TCP20 and TCP22 being the predominant interacting proteins (Supplementary Table 1). This experiment also showed that TCP20 and TCP22 could interact with both LWD1 and LWD2; thus TCP20 and TCP22 were chosen for in-depth functional studies. The reduced, but not abolished, expression of *CCA1* in *tcp20 tcp22* mutant later prompted us to explore whether additional TCPs were working with LWD proteins for the activation of *CCA1*. To test this, targeted yeast two-hybrid assays were performed to examine the interaction between LWDs and TCP6, 7, 8, 9, 11, 14, 15, 16 and 23 and these results were shown in (Supplementary Fig. 11 in the revised manuscript). This targeted Y2H suggested that additional TCPs may also play activator roles toward *CCA1*. The more complex interaction and functional networks between LWDs and TCPs will require investigations beyond the scope of this current study.

-P6L160; the statement 'TCP20 and TCP22 are required for activating *CCA1*' seems a long stretch, considered *CCA1* still gets activated in the *tcp20/tcp22* double mutant, up to ~70% of wt levels. Certainly that means other factors are involved in 'waking up' *CCA1* independent of TCP20/22.

Response:

Indeed the original description was inappropriate, we have revised the statement (P6L167) to “.....TCP20 and TCP22 are required for the full activation of *CCA1* by light”.

-The results show that TCP20 and TCP22 are both required to activate *CCA1* (i.e. the double mutant has identical effects as each single mutant). How does that match with the results in Figure 3b where only TCP20 is induced? How are the effects of overexpression of only one of them explained (figure 5a-b)?

Response:

Two possibilities might explain that overexpression of TCP20 or TCP22 alone could activate the *CCA1*. First, taking the overexpression of TCP20 as an example, if residual TCP22 monomers exist or the concentration of TCP22 is limited in the wild-type plant cells, the bimolecular interaction between TCP20 and TCP22 may be facilitated by the increased concentration of TCP20 to form a TCP20-TCP22 heterodimer. Second, although the comparison of *tcp* single and double mutants suggested that TCP20 and TCP22 function as heterodimers, one cannot entirely rule out that TCP20/TCP22 may function as homodimers, especially when the concentration is high, to activate *CCA1*.

-Modelling using both a new simple model as well as an integrated complex model suggests that it is feasible that LWD1 positively regulates both *CCA1* and its repressor PRR9. The rest of the manuscript suggests that TCP20/TCP22 can positively and negatively regulate *CCA1* expression depending on the presence of LWD1. Given the complexity, a mathematical model would really help at the end of the paper to assess whether the full set of results, including dynamics of LWD1/TCP20/TCP22, actually

explains clock behaviour. This bit of modelling could complement the visual model in Figure 5e.

Response:

We agree that given the complexity of the circadian clock, it is appealing to assess/explore the dynamics of LWD1/TCP20/TCP22 with a modeling approach. However, this will require additional information about how other clock genes interact with TCP20/TCP22 and expression analyses of *TCP20/TCP22* in other clock mutants and expression of additional clock genes in *tcp20 tcp22* mutant. This line of study certainly would be of great interest but requires additional expression assays to construct a trustworthy model including TCP20/TCP22.

Our original Figure 5e may have misled the reviewer in concluding a negative role of TCP20/TCP22 on *CCA1* in the absence of LWD1. Our data from Figure 5b and 5c only supported that the concomitant binding of LWD1 and TCP20/TCP22 is required for the full activation of *CCA1*. We have modified Figure 5e to better reflect this point.

Minor points:

-The modelling in Figure 1 only involves LWD1; are dynamics affected by including LWD2? If not, should all labels possibly read 'LWD1/2'?

Response:

Indeed, LWD1 and LWD2 have greater than 90% amino acid sequence similarity and have been shown to be functionally redundant (Wu et al., 2008). In the revised manuscript, we have replaced LWD1 with LWD1/2 in Fig. 1a.

-P2L29, first sentence of abstract states that 'clock is driven' by *CCA1/LHY* and *TOC1*. This statement lacks nuance, when knock-outs of these genes are still rhythmic.

Response:

As suggested, we have toned down the statement as “....clock is regulated by...” (P2L29).

-P3L70, sentence starting 'Here, we...' needs rewriting to improve clarity.

Response:

As suggested, we have revised the description in this paragraph to improve clarity (P3L69).

-P4L83, title needs rewriting: LWD1 does not use mathematical modelling to activate *PRR9* and *CCA1*.

Response:

Indeed. We thank the reviewer for pointing out this. In the revised manuscript, we have modified the subtitle to “Mathematical modeling supports the dual activation of *PRR9* and *CCA1* by LWD1” (P4L80).

-P4L94, statement about two parameters to search for needs clarification.

Response:

The two parameters are described in Supplementary Note P3L71-73 for clarification.

-P5L142, the data in Figure 2c would benefit from a little more explanation than one single sentence.

Response:

As suggested, we have better described the results for Fig. 2c.

-P6L160, sentence starting 'We next...' needs rewriting to improve clarity.

Response:

As suggested, we have improved the sentence in P6L167-170 in the revised manuscript.

-P7, section on 'feedback loop'; I always assumed interactions within the full feedback system are called a 'loop' when the interactions both ways are direct? Data clearly show *CCA1* does not directly bind the *TCP20/22* promoters, therefore there is no loop between *CCA1* and *TCP20/22*?

Response:

Indeed the direct binding is only for TCPs on *CCA1* promoter. In the revised manuscript, we have used more precise statements to describe the mutual regulation of TCPs and the morning genes (P6L157 and P8L216).

-*CCA1* and *LHY* are partially redundant. Could you briefly discuss on the relevance of a repressor/activator complex that only acts on one of the two?

Response:

For two functionally redundant genes such as *CCA1* and *LHY* in the circadian clock, their differential regulation may increase the flexibility and tunability of the circadian clock. We have added a brief discussion as suggested in P8L246-248.

Reviewer #3:

In this paper by Wu et al, entitled "LWD-TCP complex wakes up the morning gene CCA1 in Arabidopsis," the authors define new interactions and roles for the LWD proteins and the TCP transcription factors. *lwd1/2* mutants have a major role maintaining the amplitude and period of the circadian clock. The core-clock transcription factor CHE, which is also a TCP transcription factor, negatively regulates the amplitude of the CCA1. The authors have identified a novel, direct interaction between TCP20 and TCP22, and with LWD1 or LWD2. Through modifying and extending an ODE-based mathematical modeling of the clock (Pokhilko et al. 2012), the authors predict a co-activator role for LWD1 at the CCA1 promoter, and suggest that the LWDs are recruited to the promoters through an interaction with TCP20 and 22. TCP20 and 22 can directly bind to the CCA1 promoter in an LWD1/2 independent manner. However, LWD1 recruitment the CCA1 promoter does not require TCP20 or 22, which is likely due to redundancy in the type 1 TCP family, which broadly interact with LWDs. These findings identify the first transcriptional activators identified for CCA1. The paper is well written, data is clear and presented consistently. However, I have the following concerns:

Major concerns:

1. The authors show indirectly that LWD1, LWD2, TCP20 and TCP22 are necessary for activation by through the use of bioluminescence reporters in vivo, yet the authors do not show a complementation experiment where they observe rescue of the loss of amplitude of the CCA1 reporter. The authors should show that the loss of amplitude of the reporter is due to absence of the gene by complementing with a transgene for TCP20 or TCP22 into *tcp20* or *tcp22*, respectively. The reviewer is concerned that the loss of amplitude is due to co-suppression of the CCA1::LUC reporter by TDNA insertions in the mutant backgrounds, and not necessarily reflective of amplitude of CCA1 (Gao Y, Zhao Y. Epigenetic Suppression of T-DNA Insertion Mutants in Arabidopsis. Molecular Plant. 2013;6(2):539-545. doi:10.1093/mp/sss093). If the effect on amplitude of the reporter is direct, then amplitude of the *tcp 20* CCA1:LUC line should be rescued by complementing with a TCP20 transgene.

Response:

As suggested, we have included the results of complementation experiments in Supplementary Fig. 5 of the revised manuscript. Briefly, by introducing the TCP20 or TCP22 driven by the native promoters into the corresponding *tcp* mutants, the low amplitude of *pCCA1::LUC2* expression could be rescued to different degrees in different complementation lines. This result indicated that the reduced CCA1 in *tcp* mutants did not result from co-suppression of CCA1::LUC2 reporter in *tcp* mutants. In addition, the early-flowering phenotypes of the *tcp20-2* and *tcp22-1* mutants can be complemented by TCP20 and TCP22, respectively. These results confirmed that TCP20 and TCP22 indeed play positive roles in CCA1 expression and in photoperiodic pathway.

2. The authors suggest that concomitant recruitment of LWD1 to promoters with TCP will lead to activation. In figure 4c, TCP20 and TCP22 can induce the expression of a TBS

reporter in a transient transfection assay. If LWD is important for co-activation by TCPs, then co-transfection of LWD1 in similar experiments should lead to increased expression of the reporter. Alternatively, if endogenous LWD is a concern, then the authors could show that transfection of effectors and the reporter into *lwd1/lwd2* protoplasts leads to lower transactivation compared to wild type. This would show directly in a transient assay that LWDs can act as co-activators in concert with the TCPs through the TBS.

Response:

As suggested, we examined the effect of LWD1 as a co-activator in protoplast transient assays. Results in Supplementary Fig. 10 of the revised manuscript indicated that the trans-activation activity of TCP20 on TBS was lower in protoplasts from *lwd1 lwd2* than wild-type plants. This new result also supported that the concomitant presence of LWDs and TCPs are important for the activation of TBS on the *CCA1* promoter.

Minor concerns:

1. In figure 5e, the model, the authors suggest that LWD1 and TCP20/22 are recruited to the *CCA1* promoter directly to activate transcription, and without LWD1 and LWD2 TCPs cannot induce *CCA1*, and as drawn, TCP20/22 directly recruits LWD1. However, LWD1 can be ChIPed at the promoter in the absence of TCP20/22 similarly as with (or better maybe). Likewise, *CCA1* is still expressed in the *lwd lwd2* background, albeit with reduced amplitude and period, so the sections on the left are a little misleading. The model should better reflect the data.

Response:

We thank the reviewer for pointing out the limitation of the illustration. In the revised manuscript, we have simplified the illustration to focus on only data shown in this study.

2. In figure 4c does "Vector" in the graph refer to mini35S in the right schematic or a mock control where the TBS reporter is transformed without co-expressing TCP20? A mock control is required to determine if the TBS vector is strongly induced without addition of the TCP effector in protoplasts. A TBS alone control without a TCP effector would be an appropriate control for this experiment, and the experiment performed in supplemental figure 8.

Response:

We apologize for the inconsistency of the labeling. The "vector" indeed meant mini35S. The labeling was corrected in the revised manuscript. The mock control with TBS only but without TCP as an effector had basal level of reporter gene expression. One such example is shown in Supplementary Fig. 10 of the revised manuscript. For better clarity, results comparing TBS and mTBS/mini35S are shown in Fig. 4c and Supplementary Fig. 11c (originally Supplementary Fig. 8c) of the revised manuscript.

3. Table 2, supplemental data: typo- Dimmension should be spelled Dimension

Response:

The spelling has been corrected in the revised manuscript.

Reviewer #4:

The authors will recognize some of these comments as I reviewed an earlier version at another journal. The manuscript is improved since I last saw it. Some of my major concerns from that previous submission have been addressed and I think the manuscript provides good evidence for their major arguments that the LWD transcriptional regulators regulate the circadian oscillator gene CCA1 through the TCP class of transcription factor. There are still some areas of concern that remain. If those are unchanged from the previous version I have used the same text as the last time I saw this work.

The authors address an area of interest by investigating the the activators of LHY/CCA1 expression at dawn. The authors use a mix of modeling, yeast 2 hybrid, mutant studies, transient activation and CHIP to test the roles of LWD1/2.

I have most concerns regarding the modeling. The results section starts with a paragraph describing a modeling approach which the authors describe as comparing the potential for LWD1 to regulate PRR9 alone with models in which LWD1 regulates both PRR9 and CCA1 together. This is a misleading description because in the model CCA1 and LHY, and PRR7 and PRR9 are collapsed each in to single genes, and therefore the model compares the effect of LWD1 regulating both PRR9/PRR7 or with LWD1 regulation all of PRR9, PRR7, CCA1 and LHY. Based on the authors own data it would seem that particularly in the context of LWD1 this simplification of the model is inappropriate and makes the modeling predictions unhelpful in the context of the work presented in this manuscript. The authors have now added a re-analysis of the Pokhilko 2012 model which is a much better tool for their investigation. The remodeling in the Pokhilko model described in the second paragraph supports the conclusion that LWD1 regulates PRR9 and not PRR7.

A CHIPseq identified an interaction between LWD1 and the CCA1 promoter at ZT21, around the time when transcriptional activation of CCA1 begins. Because LWD1 has no DNA binding motif the authors looked for potential interacting proteins using yeast two hybrid and identified four potential interactors of the class I TCP transcriptional regulators. One of these, TCP21/CHE has been shown previously to repress CCA1 expression. A class II TCP3 was also found as a potential interactor. The dominant clones sequenced were TCP20 and 22, which prompted the authors to investigate these further. LWD2 was also shown to interact with TCP20/22. BiFC was used to confirm the interaction with LWD1. The TCP20, TCP22 single and double mutants had short period circadian rhythms and a short day flowering phenotype.

The expression levels of CCA1 were found to be lower in TCP20 TCP22 double mutants. The authors used transient induction experiments to demonstrate that TCP20 is an activator of CCA1. The tcp 20 and tcp 22 mutants had a slight short period phenotype.

The promoters of TCP20 and 22 oscillated under continuous light and under light and dark cycles. The authors provide evidence that the expression of TCP20 and TCP22 are under circadian control. The authors show that CCA1 and LHY do not directly repress TCP20 and TCP22.

The TCP binding site (TBS) is present only in the promoter of CCA1 and not LHY and the authors found that the TBS was bound by TCP20 and TCP22. The authors provide good evidence that TCP20 and TCP22 bind and regulate the CCA1 and not the LHY promoter

I find the conclusion that LWD1 and LWD2 are required for the induction compelling based on the TCPox studies in the *lwd1 lwd2* mutant background. Perhaps surprisingly, TCP20 and TCP22 bound the CCA1 promoter in *lwd1 lwd2* null backgrounds which lead the authors to conclude that LWD are transcriptional co-activators of TCP20 and TCP22, which I think is reasonable interpretation of the data.

Major Comments

As I have mentioned above. I am was very concerned by the modelling. approach when I received this manuscript the first time. That modeling approach remains and is described in the first paragraph of the results but it is now supplemented by a second approach which is described in lines 99 onwards.

I wrote about the modelling. approach that is described in the fist paragraph " The approach was to use a reduced model form considering LHY/CCA1 as a single gene and PRR9/PRR7 also as a single gene. The goal was to understand the function of LWD1. Since LWD1 binds only CCA1 and not LHY, and LWD1 regulates PRR9 but apparently not PRR7, it seems to me that this is a simplification too far. The experimental data demonstrate that the assumption that CCA1/LHY and PRR7/PRR9 can be considered as functional equivalents is false. For some studies the assumption, whilst false, might not produce misleading results, however in this context the goal is to understand the potential functions of LWD1, which regulates only one of each of these two gene pairs, it might be that the simplification of the model must inevitably obscure the likely roles of LWD1. Therefore almost any predictions from the model concerning LWD1 have a high probability of being false. I can not see any justification for treating these genes pairs as single genes when it is known that they are targeted differently by LWD1 and it was this regulation that the model was specifically designed to investigate.

Response:

We thank the reviewer for being willing to read our work and offer advice again. In fact, we agreed with the comments made by the reviewer in the previous review that we should NOT collapse CCA1/LHY or PRR9/PRR7 in the modeling. The manuscript submitted to Nature Communications adopted models including only LWD1/LWD2, CCA1, PRR9, for both Model I and Model II (Fig. 1a). We first inferred the positive role of LWD1/LWD2 on both *PRR9* and *CCA1* from this simplified model. Of course there may exist inherited limitations when a conclusion was drawn based on a simplified model.

That was why we further validated this conclusion by the successful integration of this regulation into a more “complete” model, the Pokhilko model (Supplementary Fig. 1), as the reviewer pointed out in this new manuscript.

The explanation of the model is far from clear "among 2.4x10⁸ random parameter sets examined, 1,004 sets met our criteria (see Methods) for Model II as compared with only 27 of 3x10⁸ random parameter sets for Model I." What criteria for what? And "Thus, faithful oscillation was greater than 40 times easier to obtain when LWD1 also activates CCA1, despite having two more parameters to search for in Model II. " What is meant by a faithful oscillation, how can it be 40x easier to obtain? This all needs much better explanation. All my criticisms concerning this aspect of the manuscript remain.

Response:

After receiving the reviewer’s comments from the previous submission, we have in fact described the criteria we chose in the Supplementary Note (P2L57-P3L66) of the submission to Nature Communications and have improved the description in results, but those may still be insufficient. For better clarity, we therefore have elaborated further in the results (P4L88-100) and also the criteria used in the Supplementary Note (P2L53-P3L65).

Briefly, all of the 9 independent parameters were obtained by random search, propagate, and screened for regular oscillation, except for the Hill coefficients (n), which are fixed at 3; the search was performed at a logarithmic scale across three orders of magnitude, for γ 's and κ 's, and a linear scale for α 's. Each parameter was varied by their minimum or maximum values as shown in Table 1 in the Supplementary Note. The criteria we used were as follows:

- (1) The trajectory must oscillate regularly, which was defined by examining the period and amplitude change in each cycle. We calculated the relative difference of period and amplitude for each cycle, defined as $|(x_1-x_2)|/\min(x_1, x_2)$, where x_1 and x_2 are the period or amplitude calculated from two consecutive cycles. An acceptable regular oscillation has less than 5% relative change for more than 10 cycles.
- (2) In the *lwd1 lwd2* mutant, the oscillation must have reduced amplitude (>50%) and shorter period (<21 h), as reported previously (The Plant Cell, 2011, 23:486).
- (3) To avoid nonphysical sensitivities to small changes in the simulation, the parameter set must generate similar results from two different ODE solvers (ODE15s and ODE23s).

The authors have improved on the original version by adding a new part to the modelling. in which they examine the role of LWD1 in the Pokhilko model (starting at line 99 of the MS). The is a more sensible strategy than the modelling. described in the first paragraph of the MS. The manuscript would be improved by removal of the first modeling strategy. I do not think it provides meaningful insight, whereas the use of the Pokhilko model is useful.

Response:

With the simplified models, we aim to gain insights for the plausible role of LWD1/LWD2. With the conclusions obtained from the simplified models, we further asked whether the circadian dynamics can be maintained if the new interactions are added to a more complex model. The simplified model, with the new interaction between LWD1 and *CCA1*, much easier to form regular oscillations and is able to reproduce short period lengths for *cca1* and *toc1* mutants and long period length for *prr9*. Such a conclusion cannot be easily replaced by an extended Pokhilko model, because the Pokhilko model consists of only 1 parameter set that works very well, whereas for the simplified models, we searched for and obtained many plausible parameter sets and concluded the probability of obtaining such parameter sets. The same conclusion would have been obtained from the Pokhilko model if we had searched all the parameter space and required the model to reproduce the fitting of massive experimental data. However, in doing so, the probability of obtaining a good parameter set with a random search would have been unachievably low. Moreover, in the complex model, there are many possible compensations for the effects of LWD1/LWD2, and mutations in *LWD1/LWD2* may not lead to a significant difference. Therefore, we used a simple model to draw general insights and a more complex model to show that LWD1/LWD2 can be added and still maintain the dynamics of circadian oscillations.

In this work, we show that although the model was simple, the conclusion we obtained was still valid when extending to the more complex model. The primary advantage of using a simplified model is the high efficiency in searching the limited number of parameters extensively. In a simplified model, the roles of the components of interest are less likely to be compensated for by other components in a more complex model. For example, the Pokhilko model can perform well without including LWD1/LWD2, which implies that the effects of LWD1/LWD2 maybe indirectly included or compensated for inside the complicated network structures and the parameters. Therefore, we feel the results from the simplified model we adopted are valuable and were shown to be very helpful in revealing the dual activating functions of LWD1 on *PRR9* and *CCA1* in the current work.

The authors still do not explain why some of the experiments were performed in constant dark (DD). To interpret the data it needs to be explained why the experiments were performed in constant dark.

Response:

Experimental results shown in Fig. 3a were collected from plants grown in constant dark. In the revised manuscript, we have elaborated the experimental design in P6L159-164.

The authors now report that *TCP20* and *22* mutants have a very small phenotype of slightly shorter than wild type

Response:

Although not dramatic, *tcp* mutants have short period length and early-flowering phenotypes, thereby supporting *TCP20/TCP22* functioning in the regulation of the circadian clock and photoperiodic flowering pathway. Functional redundancy among

class I TCPs may explain why the *tcp20 tcp22* double mutant does not have more dramatic phenotypes as described in P9L267-279.

In my review of a previous version of the manuscript I made negative comments about the figure legends. These comments still apply.

Figure 1 the legend is unclear. How can there be period estimate differences with errors for the deterministic models? Surely deterministic models will always produce the same period estimate? If these were the means of the different simulations, how is this informative when a huge range of parameters were used? More explanation is needed.

Response:

The periods in Fig. 1a were estimated using different parameter sets. In this figure, we used a good majority of the parameter sets obtained that give rise to clock oscillation after genetic perturbation from both Models I and II. Because the effects of this genetic perturbation depend on the combination of parameter values in the set, each parameter set has a different period in the perturbation, and therefore the figure shows the distribution of the period lengths of multiple parameter sets after genetic perturbation. The legend of Fig. 1a has been improved in the revised manuscript. Furthermore, we have improved the Supplementary Note to explain how the genetic perturbation test was conducted (Supplementary Note P3L77-P4L92).

This set of results shows that, even though the parameter values were searched from a large parameter space, most of the parameter sets obtained have a similar and correct behavior in all the genetic perturbation tests we performed for Model II but not for Model I. Such general property of the simple models allows us to conclude that the structure of the model, not the detail of the parameter settings, has desirable properties in mimicking the clock.

Why were amplicons a and b chosen, what is their significance?

Response:

Amplicons 'a' and 'b' cover ~1-kb promoter region of *CCA1* and were selected for ChIP experiments according to previous study (The Plant Cell, 2011, 23: 486). Amplicon 'b' also harbors a TCP-binding site (TBS), which allowed us to test for the direct and specific binding of TCPs to this region.

Figure 2 the legend is too incomplete to understand the data. Specifically what is being shown? The authors could expand the figure legend and still comply with the article length restrictions by reducing the length of the final conclusion which is a bit repetitive of the text in the rest of the manuscript.

Response:

As suggested, we have provided more information in the legend of Fig. 2 for clarity.

Minor Comments

I criticized the abstract in my review of a previous version of the paper. Those criticisms still apply

The abstract contains many inaccuracies and over simplifications of the circadian system which are misleading. Such as the following which I believe to be incorrect "CCA1 initiates and sets the circadian phase, which requires its peak expression at dawn." The phase is a result of many circadian components and CCA1 is usually considered to peak shortly after dawn. And the Arabidopsis circadian clock is driven by a double negative feedback loop formed by the morning genes CCA1/LHY and the evening gene TOC1". I disagree, these are only part of the system.

Response:

The reviewer's points are taken. We have toned down the statements in the abstract in the revised manuscript.

Diurnal is used incorrectly in this manuscript. Diurnal is the antonym of nocturnal and should not be used to describe light and dark cycles.

Response:

As suggested, we have replaced the "diurnal cycle" with "light/dark cycles" for accuracy.

The method still do not state what media the plants were grown on. This is essential information

Response:

As suggested, we have clearly described the plant media in the methods section of the revised manuscript.

It was not clear to me if LWD2 interacted with the same members of the TCP clade as LWD1. Is this described anywhere? Apologies if I missed those data. It would be good to make clear whether those interactions were tested, and if they were, what the result was. If they were not tested I think that is OK, but this needs to be clear.

Response:

Our yeast two-hybrid results indicated that LWD1 and LWD2 showed similar interaction affinities with TCP family members examined. Interaction pairs tested were marked in Fig. 2a and Supplementary Fig. 11 of the revised manuscript.

REVIEWERS' COMMENTS:

Reviewer #1 (Remarks to the Author):

I am basically happy that authors answered my questions properly. But how authors revised the ms in response to my first comment, "whether newly found components in this study and its regulatory pathways play an important role for clock function is not clear", was not explained.

Reviewer #2 (Remarks to the Author):

Wu et al., NCOMMS-16-08634B

Dear authors,

I am satisfied that the comments I made in the original review have been addressed appropriately, and I now support publication of this manuscript.

This was an open review by Gerben van Ooijen.

Reviewer #3 (Remarks to the Author):

In this revised manuscript by Wu et al, entitled "LWD-TCP complex wakes up the morning gene CCA1 in Arabidopsis," the authors define new interactions and roles for the LWD proteins and the TCP transcription factors. *lwd1/2* mutants have a major role maintaining the amplitude and period of the circadian clock. The core-clock transcription factor CHE, which is also a TCP transcription factor, negatively regulates the amplitude of the CCA1. The authors have identified a novel, direct interaction between TCP20 and TCP22, and with LWD1 or LWD2. Through modifying and extending an ODE-based mathematical modeling of the clock (Pokhilko et al. 2012), the authors predict a co-activator role for LWD1 at the CCA1 promoter, and suggest that the LWDs are recruited to the promoters through an interaction with TCP20 and 22. TCP20 and 22 can directly bind to the CCA1 promoter in an LWD1/2 independent manner. However, LWD1 recruitment the CCA1 promoter does not require TCP20 or 22, which is likely due to redundancy in the type 1 TCP family, which broadly interact with LWDs. In transient assays, activation of reporters by TCP20 require LWD1 and LWD2. These findings identify the first transcriptional activators identified for CCA1. The paper is well written, data is clear, presented consistently, and the authors have adequately addressed my comments by new experiments, adding or editing text, and updating figures where necessary.

Reviewer #4 (Remarks to the Author):

I was concerned by the collapsing of CCA1/LHY in to a single gene and PRR7/9 in to a single gene when the purpose is to show specific effects of LWD1/2 on specifically CCA1 and not LHY. The authors argue that model presented at the beginning of the manuscript is is not collapsing the model to single genes, but now call it a simplified model. However I can not see how this is any different from describing the model as collapsing the genes in to one gene. I might be mistaken but it would appear to me that the author's reply is not more than semantics. Whether one describes CCA1 and LHY being collapsed to one gene called CCA1, or one makes a "simplified model" with only CCA1 present, the end point is the same, because LHY and PRR7 are not present in the loops in which the specific effects on their partner genes are being investigated I see little value in this model. I also can not see how the

model is different, the authors response indicate that is different and not a collapsed model but a simplified model, to this reader it seems that only the description has changed but this essentially means the same thing. I do not think the model presented in Figure 1 can represent a realistic model of the specific effects of LWD1/2 on PRR9 and CCA1 when their partner genes PRR7 (for PRR9) and LHY (for CCA1) are not included. This makes interpretation of the model very challenging since if LWD1/2 are affecting only one gene in each half of the loop does this have an overall effect? Nevertheless, the authors in their response concerning the extension to the Polikho model make a very good point that the findings of the simple model allowed them to examine in the very complex Polikho model that would not have been possible without the guidance of the simplified model. I suggest that the authors revise the manuscript to be very explicit about the limitations of the simplified model and explain how it was essential to guide them in the examination of a more complex model. This would be a more open and informative way of describing the modeling. Most of my other comments have been resolved only the following remain outstanding

"Why were amplification a and b chosen, what is their significance? Response: amplification 'a' and 'b' cover ~1-kb promoter region of CCA1 and were selected for ChIP experiments according to previous study (The Plant Cell, 2011, 23: 486). Amplicon 'b' also harbors a TCP-binding site (TBS), which allowed us to test for the direct and specific binding of TCPs to this region." Is this information included in the new manuscript? It should be included.

I criticized the abstract. My criticism and the authors' response are pasted below.

"The abstract contains many inaccuracies and over simplifications of the circadian system which are misleading. Such as the following which I believe to be incorrect "CCA1 initiates and sets the circadian phase, which requires its peak expression at dawn." The phase is a result of many circadian components and CCA1 is usually considered to peak shortly after dawn. And the Arabidopsis circadian clock is driven by a double negative feedback loop formed by the morning genes CCA1/LHY and the evening gene TOC1". I disagree, these are only part of the system. Response: The reviewer's points are taken. We have toned down the statements in the abstract in the revised manuscript."

Despite the authors' response we can see from the abstract, pasted below, the sentences I criticize have not been modified. "The Arabidopsis circadian clock is regulated by a double-negative feedback loop formed by the morning genes CIRCADIAN CLOCK ASSOCIATED1 (CCA1)/LATE ELONGATED HYPOCOTYL (LHY) and the evening gene TIMING OF CAB EXPRESSION1 (TOC1)."

The authors have satisfactorily answered my question concerning why the experiments were performed in DD. It was not clear that there is a light pulse at T = 24, and this needs to be added to the figure in this experiment and elsewhere if it is needed.

Responses to the reviewers' comments

Reviewer #1:

I am basically happy that authors answered my questions properly. But how authors revised the ms in response to my first comment, "whether newly found components in this study and its regulatory pathways play an important role for clock function is not clear", was not explained.

Response: We apologize for not responding to this comment in the previous revision. Our study has demonstrated TCP20 and TCP22 could control the flowering time by regulating the circadian period length and by activating the expression of the morning gene *CCA1*. We thus consider the identification and mechanistic studies of TCP20 and TCP22 has expanded the current understanding of clock function and operation in *Arabidopsis*.

Reviewer #2:

Dear authors,

I am satisfied that the comments I made in the original review have been addressed appropriately, and I now support publication of this manuscript.

This was an open review by Gerben van Ooijen.

Response: We thank Dr. Gerben Van Ooijen for his valuable comments.

Reviewer #3:

In this revised manuscript by Wu et al, entitled "LWD-TCP complex wakes up the morning gene *CCA1* in *Arabidopsis*," the authors define new interactions and roles for the LWD proteins and the TCP transcription factors. *lwd1/2* mutants have a major role maintaining the amplitude and period of the circadian clock. The core-clock transcription factor CHE, which is also a TCP transcription factor, negatively regulates the amplitude of the *CCA1*. The authors have identified a novel, direct interaction between TCP20 and TCP22, and with LWD1 or LWD2. Through modifying and extending an ODE-based mathematical modeling of the clock (Pokhilko et al. 2012), the authors predict a co-activator role for LWD1 at the *CCA1* promoter, and suggest that the LWDs are recruited to the promoters through an interaction with TCP20 and 22. TCP20 and 22 can directly bind to the *CCA1* promoter in an LWD1/2 independent manner. However, LWD1 recruitment the *CCA1* promoter does not require TCP20 or 22, which is likely due to redundancy in the type 1 TCP family, which broadly interact with LWDs. In transient assays, activation of reporters by TCP20 require LWD1 and LWD2. These findings identify the first transcriptional activators identified for *CCA1*. The paper is well written, data is clear, presented consistently, and the authors have adequately addressed my comments by new experiments, adding or editing text, and updating figures where necessary.

Response: We thank this reviewer for recognizing our efforts in improving the previous version.

Reviewer #4 (Remarks to the Author):

I was concerned by the collapsing of CCA1/LHY in to a single gene and PRR7/9 in to a single gene when the purpose is to show specific effects of LWD1/2 on specifically CCA1 and not LHY. The authors argue that model presented at the beginning of the manuscript is is not collapsing the model to single genes, but now call it a simplified model. However I can not see how this is any different from describing the model as collapsing the genes in to one gene. I might be mistaken but it would appear to me that the author's reply is not more than semantics. Whether one describes CCA1 and LHY being collapsed to one gene called CCA1, or one makes a "simplified model" with only CCA1 present, the end point is the same, because LHY and PRR7 are not present in the loops in which the specific effects on their partner genes are being investigated I see little value in this model. I also can not see how the model is different, the authors response indicate that is different and not a collapsed model but a simplified model, to this reader it seems that only the description has changed but this essentially means the same thing. I do not think the model presented in Figure 1 can represent a realistic model of the specific effects of LWD1/2 on PRR9 and CCA1 when their partner genes PRR7 (for PRR9) and LHY (for CCA1) are not included. This makes interpretation of the model very challenging since if LWD1/2 are affecting only one gene in each half of the loop does this have an overall effect? Nevertheless, the authors in their response concerning the extension to the Polikho model make a very good point that the findings of the simple model allowed them to examine in the very complex Polikho model that would not have been possible without the guidance of the simplified model. I suggest that the authors revise the manuscript to be very explicit about the limitations of the simplified model and explain how it was essential to guide them in the examination of a more complex model. This would be a more open and informative way of describing the modeling.

Response: The reviewer might have mistaken that we simplified the model with the purpose of showing specific effects of LWD1/2 on CCA1, but not LHY. In fact, the model was simplified because we agreed with the reviewer that it was inappropriate to collapse CCA1 and LHY (and also PRR9 and PRR7) in our earlier mathematical modeling. And, our current results indicated that LWD-TCP complex regulates only CCA1, but not LHY (Supplementary Fig. 8), further supported collapsing CCA1 and LHY could be misleading.

We previously argued that the primary advantage of using a simplified model is the high efficiency in searching the limited number of parameters in an extensive way. To begin with a simplified model, we could search for and obtain many plausible parameter sets, compared to the more complex Pokhilko model that consists of only 1 parameter set that works effectively. Also, compared to a more complex model with multiple components, the roles of a specific component are less likely to be compensated for by the others. For example, the dual activating roles of LWD1/2 on PRR9 and CCA1 could be clearly revealed by the simplified model, whereas the roles of LWD1/2 could be compensated for or indirectly included in the Pokhilko model.

As suggested by the reviewer, we have indicated the possible limitation of using the simplified model (page 4 line 99) in the revised manuscript. Our attempt in addressing the possible limitations by integrating our findings to the more complex model has been described in the manuscript (now page 4 line 100 to page 5 line 118) and also in the Supplementary Note 1.

Most of my other comments have been resolved only the following remain outstanding "Why were amplification a and b chosen, what is their significance? Response: amplification 'a' and 'b' cover ~1-kb promoter region of CCA1 and were selected for ChIP experiments according to previous study (The Plant Cell, 2011, 23: 486). Amplicon 'b' also harbors a TCP-binding site (TBS), which allowed us to test for the direct and specific binding of TCPs to this region." Is this information included in the new manuscript? It should be included.

Response: Fragment b contains TBS was described in page 8 lines 222-223 in the revised manuscript.

I criticized the abstract. My criticism and the authors' response are pasted below.

"The abstract contains many inaccuracies and over simplifications of the circadian system which are misleading. Such as the following which I believe to be incorrect "CCA1 initiates and sets the circadian phase, which requires its peak expression at dawn." The phase is a result of many circadian components and CCA1 is usually considered to peak shortly after dawn. And the Arabidopsis circadian clock is driven by a double negative feedback loop formed by the morning genes CCA1/LHY and the evening gene TOC1". I disagree, these are only part of the system. Response: The reviewer's points are taken. We have toned down the statements in the abstract in the revised manuscript."

Despite the authors' response we can see from the abstract, pasted below, the sentences I criticize have not been modified. "The Arabidopsis circadian clock is regulated by a double-negative feedback loop formed by the morning genes CIRCADIAN CLOCK ASSOCIATED1 (CCA1)/LATE ELONGATED HYPOCOTYL (LHY) and the evening gene TIMING OF CAB EXPRESSION1 (TOC1)."

Response: We certainly agree the loop of CCA1/LHY and TOC1 is only part of the system. However, we do not think our original statement is exclusive or implies CCA1/LHY and TOC1 are the only regulators of *Arabidopsis* circadian clock. To make it more explicit, we now modify the sentence to "The *Arabidopsis* circadian clock is primarily regulated by a double-negative feedback loop....".

The authors have satisfactorily answered my question concerning why the experiments were performed in DD. It was not clear that there is a light pulse at T = 24, and this needs to be added to the figure in this experiment and elsewhere if it is needed.

Response: The treatment of 1 h light at DD24 was marked in Fig. 3a and now also described in the Fig. 3a legend and text (page 6 line 161-162).